# Reconstructing lost BOLD signal in individual participants using deep machine learning

Yuxiang Yan[1,2,10], Louisa Dahmani[1,3,10], Jianxun Ren[1,4,10], Lunhao Shen[1,4], Xiaolong Peng [1], Ruiqi Wang[1], Changgeng He[1,4], Changqing Jiang [4], Chen Gong[4], Ye Tian [4], Jianguo Zhang[5], Yi Guo[6], Yuanxiang Lin[7], Shijun Li[1], Meiyun Wang[3,11✉], Luming Li[4,8,11✉], Bo Hong[2,11✉] & Hesheng Liu [1,8,9,11✉]

Signal loss in blood oxygen level-dependent (BOLD) functional neuroimaging is common and can lead to misinterpretation of findings. Here, we reconstructed compromised fMRI signal using deep machine learning. We trained a model to learn principles governing BOLD activity in one dataset and reconstruct artificially compromised regions in an independent dataset, frame by frame. Intriguingly, BOLD time series extracted from reconstructed frames are correlated with the original time series, even though the frames do not independently carry any temporal information. Moreover, reconstructed functional connectivity maps exhibit good correspondence with the original connectivity maps, indicating that the model recovers functional relationships among brain regions. We replicated this result in two healthy data-sets and in patients whose scans suffered signal loss due to intracortical electrodes. Critically, the reconstructions capture individual-specific information. Deep machine learning thus presents a unique opportunity to reconstruct compromised BOLD signal while capturing features of an individual's own functional brain organization.

[1] Athinoula A. Martinos Center for Biomedical Imaging, Department of Radiology, Massachusetts General Hospital, Harvard Medical School, Charlestown, MA, USA. [2] Department of Biomedical Engineering, School of Medicine, Tsinghua University, Beijing, China. [3] Department of Radiology, Zhengzhou University People Hospital & Henan Provincial People's Hospital, Zhengzhou, China. [4] National Engineering Laboratory for Neuromodulation, School of Aerospace Engineering, Tsinghua University, Beijing, China. [5] Department of Neurosurgery, Tiantan Hospital, Capital Medical University, Beijing, China. [6] Department of Neurosurgery, Peking Union Medical College Hospital, Beijing, China. [7] Department of Neurosurgery, First Affiliated Hospital of Fujian Medical University, Fuzhou, China. [8] Beijing Institute for Brain Disorders, Capital Medical University, Beijing, China. [9] Department of Neuroscience, Medical University of South Carolina, Charleston, SC, USA. [10]These authors contributed equally: Yuxiang Yan, Louisa Dahmani, Jianxun Ren. [11]These authors jointly supervised this work: Meiyun Wang, Luming Li, Bo Hong, Hesheng Liu. ✉email: mywang@ha.edu.cn; lilm@tsinghua.edu.cn; hongbo@tsinghua.edu.cn; hesheng.liu@mgh.harvard.edu

The blood oxygen level-dependent (BOLD) signal, acquired during functional magnetic resonance imaging (fMRI), is subject to a number of artifacts, such as magnetic susceptibility artifacts and interference from metal implants. For example, intracortical electrodes implanted in patients interfere with the BOLD signal, potentially due to their lead connectors, resulting in a significant signal loss in brain regions close to the connection site on the skull[1]. This hampers studies that investigate whole-brain activity and functional connectivity and may result in misinterpretation of findings. To date, there are no post-processing MRI methods that can mitigate such interference.

A newly proposed deep machine learning model, called deep convolutional generative adversarial networks (DCGAN), provides a possible solution for reconstructing lost information[2–6]. In the DCGAN approach, two networks—a generator and a discriminator—are pitted one against the other and are trained and optimized simultaneously. Remarkably, it does not simply assemble pieces of images it was trained on, but rather generates new images that are internally cohesive. For example, DCGAN models can successfully fill in missing portions of photographs of human faces[7] and create pictures of human faces, birds, and even art. Like photographs, BOLD images carry internally cohesive information. Embedded within resting-state data, for example, is information about BOLD signal fluctuations in each cortical surface vertex[8], from which we can extract meaningful information such as functional connectivity and task-evoked brain activity[9].

Here, we show that DCGAN can be harnessed to learn individual patterns of brain activity and generate BOLD signals in artificially and non-artificially compromised cortical regions. We trained a deep learning model on a sample of the Brain Genomics Superstruct Project (GSP) data set[10], containing intact BOLD frames from healthy young adult participants (Fig. 1a). We used the trained model to reconstruct BOLD images, frame by frame, in an independent test sample from the GSP data set, in which we

had artificially removed cortical surface regions of different sizes (Fig. 1b). Although the individual input frames did not carry information about the evolution of the BOLD signal through time, we set out on the ambitious goal of investigating the times series and functional connectivity (FC) maps extracted from the reconstructed frames. We hypothesized that the reconstructed times series and FC maps would bear high similarity to the original ones. Additionally, the large amount of resting-state data that was available enabled us to calculate individual-level functional connectivity[11,12]. We thus tested whether machine learning can be used to reconstruct individual-specific information, or whether its ability is limited to generating images based on group-level information. We replicated our analyses using the Human Connectome Project (HCP) data set[13] and compared our DCGAN reconstructions to those generated through a simple diffusion-based algorithm. Finally, we tested the DCGAN model in a clinical application, where we acquired a unique data set by collecting extensive resting-state fMRI data both before and after electrode implantation surgery in patients with Parkinson's disease. We sought to reconstruct regions in the post-operative scans that suffered substantial interference from the deep-brain stimulation (DBS) electrodes and connectors. The availability of pre-implantation scans meant we had a reference against which to compare the reconstructed images, assuming functional connectivity is stable and unchanged by the implantation surgery. We hypothesized that the reconstructed BOLD signals from the post-surgical data would be highly similar to the pre-surgical BOLD signals and that they would reflect patterns of activity that are specific to the individual.

## Results

**Reconstructed BOLD signals are correlated with original signals.** The DCGAN model was trained on a resting-state fMRI data set of 80 randomly chosen participants (240 frames for each

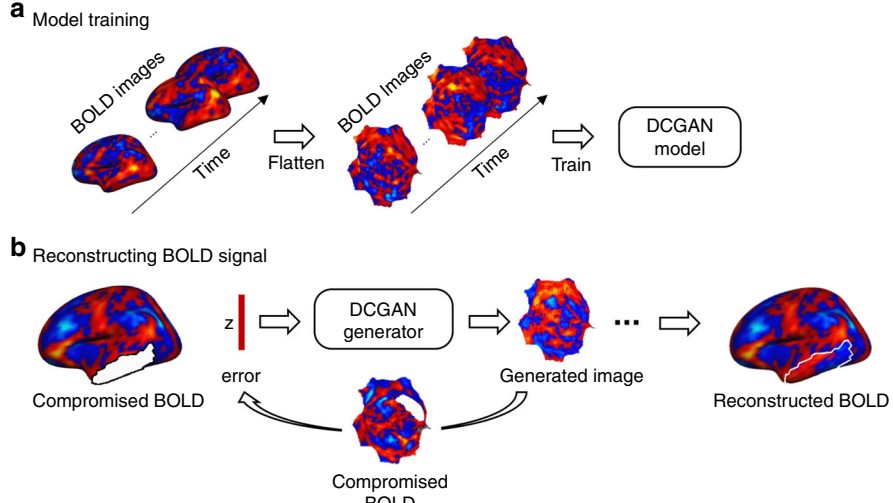

**Fig. 1 Method for reconstructing lost signal in blood oxygen level-dependent (BOLD) images using deep convolutional generative adversarial networks (DCGAN). a** The DCGAN model can be used to fill in the missing information. 19,200 BOLD frames from 80 participants are flattened and fed to the DCGAN model for training. The DCGAN model consists of two networks, a generator, and a discriminator. The generator aims to learn the distribution of the BOLD activity within the frames used as input and makes a projection from a random vector $z$, sampled from latent space $Z$, to a flattened BOLD frame $G(z)$. The discriminator is trained to distinguish the real BOLD frames from the BOLD patterns simulated by the generator. The generator and discriminator are optimized simultaneously through the two-player minimax game. **b** The trained generator is used to reconstruct the compromised BOLD frames. We first created compromised BOLD frames by removing the BOLD signal in predefined regions (here, in the temporal cortex, shown as a white mask). Each compromised BOLD frame is flattened and inputted to the trained generator. Vector $z$ is iteratively optimized by the gradient-descent method in the latent space to minimize the difference between the generated BOLD frames $G(z)$ and the authentic data $x$ outside the masked region. The vector is then projected to the flattened map $G(z_n)$ via the generator. Then, $G(z_n)$ is projected back into a 2562-vertex mesh representing the reconstructed BOLD activity in a single frame.

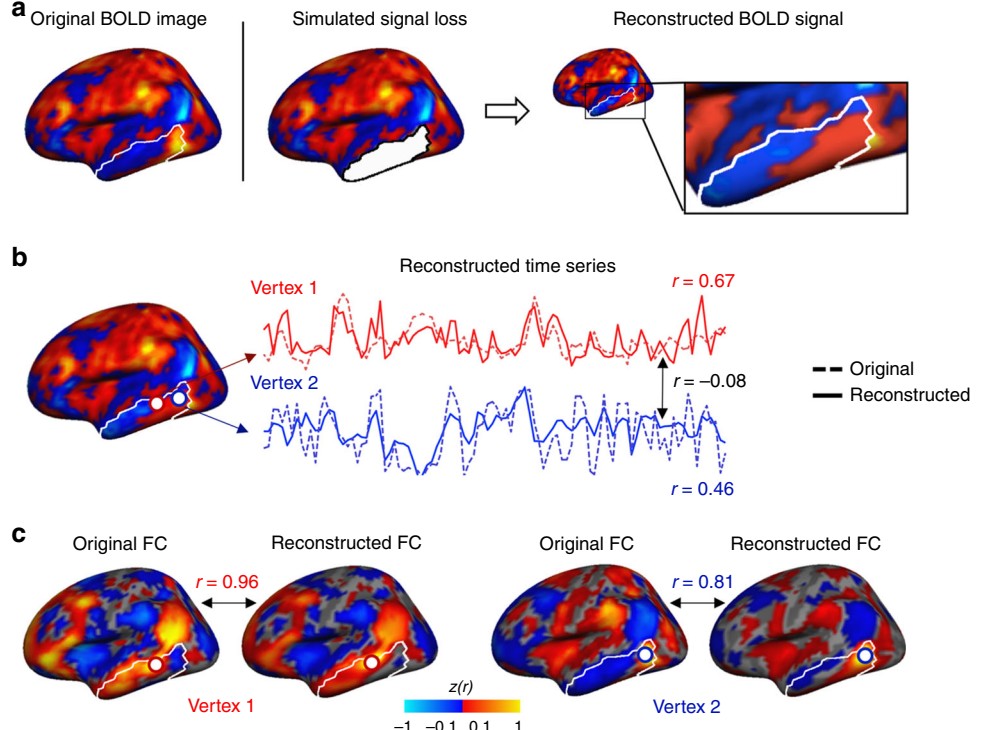

**Fig. 2 The reconstructed BOLD signals are highly similar to the original signals. a** Here we show an example of an intact BOLD frame from a representative participant in the GSP data set, which we used to create an artificially compromised BOLD frame by removing the signal in a predefined region (here, a region within the temporal cortex). The compromised BOLD frame is fed to the trained DCGAN model, which then generates a reconstructed BOLD frame based on the information within the compromised frame. **b** The time series of two vertices in the reconstructed region are shown (solid lines), along with these vertices' original time series taken from the intact BOLD frame (dashed lines). Each white circle represents a vertex. The reconstructed time series are highly similar to the original time series, as they exhibit high correlations ($r = 0.67$ for the left-most vertex outlined in red (Vertex 1) and $r = 0.46$ for the right-most vertex outlined in blue (Vertex 2)). To illustrate that the DCGAN model does not simply generate time series that follow the same fluctuations over time, we correlated the time series of the two generated vertices. The resulting correlation is $r = -0.08$, indicating that the DCGAN model takes into account the variability in activity between different vertices. **c** We investigated the functional connectivity (FC) maps of the original and reconstructed vertices. The two reconstructed FC maps show high similarity to the original FC maps ($r = 0.96$ for Vertex 1 and $r = 0.81$ for Vertex 2). These findings indicate that the DCGAN model is able to learn time-varying and functional connectivity characteristics of BOLD activity within individuals and to generate images that are highly realistic.

participant, in vertex space) from the publicly available GSP database[10]. In an independent test data set comprised of 20 participants, we artificially removed the BOLD signal in various cortical surface regions and used the trained generator to reconstruct compromised BOLD frames in vertex space.

We found that the reconstructed BOLD frames appear similar to the original intact images (see Fig. 2a for an example). To quantitatively evaluate the reconstructive accuracy of the DCGAN model, we concatenated the reconstructed frames and compared the reconstructed and original time series. This is a particularly challenging endeavor, as images are reconstructed at each time point, and each frame does not independently hold temporal information. To conduct this comparison, we assessed the correlation between the original and reconstructed BOLD time series for each vertex within each of the artificially compromised regions, located in different lobes (see "Methods" for more details), and calculated the overall average of these correlations across all participants. Table 1 shows the correlation $r$ coefficients and statistical values of this model (along with those of other models and data sets, described later). We found significant positive correlations, using multilevel linear models corrected for multiple comparisons using the Bonferroni correction: temporal cortex region: $r = 0.33 \pm 0.02$ (95% confidence intervals (CI) [0.32, 0.34], $t(19) = 65.49$, bootstrapped $p < 0.001$), lateral frontal cortex region: $r = 0.16 \pm 0.07$ (CI [0.12, 0.19],

$t(19) = 9.74$, $p < 0.001$), medial frontal cortex region: $r = 0.23 \pm 0.04$ (CI [0.21, 0.25], $t(19) = 23.24$, $p < 0.001$), parietal cortex region: $r = 0.25 \pm 0.03$ (CI [0.24, 0.27], $t(19) = 37.03$, $p < 0.001$), and occipital cortex region: $r = 0.37 \pm 0.06$ (CI [0.34, 0.40], $t(19) = 26.65$, $p < 0.001$) (Table 1). Although these correlations are low to moderate, the sheer fact that the DCGAN model was able to learn individual-specific features and capture part of the BOLD fluctuations in regions with complete signal loss is impressive. Figure 2b shows an example of a reconstruction of the lateral temporal cortex in one participant, where the correlations between the reconstructed and original time series of two randomly selected vertices are $r = 0.67$, $p < 0.001$ for Vertex 1 and $r = 0.46$, $p < 0.001$ for Vertex 2. Importantly, within the compromised region, the reconstructed BOLD signals exhibit various patterns of activity that are not necessarily correlated with each other. For example, the correlation between the time series of the two vertices above is $r = -0.08$, $p = 0.22$ (Fig. 2b).

We also assessed BOLD reconstructive accuracy by generating FC maps for all reconstructed vertices and comparing them to the FC maps of the corresponding vertices in the original intact BOLD frames. As an example, in Fig. 2c, we show the similarity between reconstructed and original FC maps for the same two temporal vertices as in Fig. 2b, in a representative participant. The FC map similarity is very high, with $r = 0.96$, $p < 0.001$ for Vertex 1 and $r = 0.81$, $p < 0.001$ for Vertex 2. In Fig. 3, we show original

**Table 1 Time series and functional connectivity map reconstruction accuracy following various models.**

| Reconstruction accuracy | r | St. dev. | t | p | Reconstruction accuracy | r | St. dev. | t | p |
|---|---|---|---|---|---|---|---|---|---|
| Time series | | | | | Functional connectivity maps | | | | |
| GSP DCGAN | | | | | GSP DCGAN | | | | |
| Temporal cortex | 0.33 | 0.02 | 65.49 | <0.001 | Temporal cortex | 0.62 | 0.06 | 47.88 | <0.001 |
| Lateral frontal cortex | 0.16 | 0.07 | 9.74 | <0.001 | Lateral frontal cortex | 0.60 | 0.04 | 65.04 | <0.001 |
| Medial frontal cortex | 0.23 | 0.04 | 23.24 | <0.001 | Medial frontal cortex | 0.61 | 0.05 | 51.87 | <0.001 |
| Lateral parietal cortex | 0.25 | 0.03 | 37.03 | <0.001 | Lateral parietal cortex | 0.70 | 0.03 | 110.62 | <0.001 |
| Occipital cortex | 0.37 | 0.06 | 26.65 | <0.001 | Occipital cortex | 0.79 | 0.05 | 68.39 | <0.001 |
| GSP (no GSR) DCGAN | | | | | GSP (no GSR) DCGAN | | | | |
| Temporal cortex | 0.33 | 0.02 | 69.29 | 0.001 | Temporal cortex | 0.61 | 0.06 | 42.19 | <0.001 |
| Lateral frontal cortex | 0.17 | 0.08 | 9.93 | <0.001 | Lateral frontal cortex | 0.62 | 0.05 | 59.56 | <0.001 |
| Medial frontal cortex | 0.23 | 0.04 | 23.64 | <0.001 | Medial frontal cortex | 0.60 | 0.06 | 46.13 | <0.001 |
| Lateral parietal cortex | 0.25 | 0.03 | 34.43 | <0.001 | Lateral parietal cortex | 0.69 | 0.03 | 99.54 | <0.001 |
| Occipital cortex | 0.37 | 0.07 | 25.33 | <0.001 | Occipital cortex | 0.78 | 0.06 | 60.96 | <0.001 |
| GSP diffusion | | | | | GSP diffusion | | | | |
| Temporal cortex | 0.24 | 0.06 | 18.20 | <0.001 | Temporal cortex | 0.51 | 0.10 | 22.54 | <0.001 |
| Lateral frontal cortex | 0.10 | 0.04 | 9.70 | <0.001 | Lateral frontal cortex | 0.40 | 0.11 | 15.85 | <0.001 |
| Medial frontal cortex | 0.17 | 0.05 | 16.11 | <0.001 | Medial frontal cortex | 0.46 | 0.09 | 22.03 | <0.001 |
| Lateral parietal cortex | 0.18 | 0.04 | 22.14 | <0.001 | Lateral parietal cortex | 0.52 | 0.05 | 47.23 | <0.001 |
| Occipital cortex | 0.29 | 0.08 | 16.46 | <0.001 | Occipital cortex | 0.59 | 0.13 | 19.42 | <0.001 |
| HCP DCGAN | | | | | HCP DCGAN | | | | |
| Temporal cortex | 0.31 | 0.05 | 27.60 | <0.001 | Temporal cortex | 0.60 | 0.07 | 38.04 | <0.001 |
| Lateral frontal cortex | 0.12 | 0.04 | 14.33 | <0.001 | Lateral frontal cortex | 0.44 | 0.05 | 39.47 | <0.001 |
| Medial frontal cortex | 0.18 | 0.05 | 16.73 | <0.001 | Medial frontal cortex | 0.46 | 0.06 | 32.53 | <0.001 |
| Lateral parietal cortex | 0.24 | 0.05 | 19.71 | <0.001 | Lateral parietal cortex | 0.62 | 0.03 | 81.92 | <0.001 |
| Occipital cortex | 0.35 | 0.07 | 23.39 | <0.001 | Occipital cortex | 0.65 | 0.06 | 46.90 | <0.001 |
| HCP (temporally smoothed) DCGAN | | | | | HCP (temporally smoothed) DCGAN | | | | |
| Temporal cortex | 0.31 | 0.05 | 27.32 | <0.001 | Temporal cortex | 0.62 | 0.06 | 43.39 | <0.001 |
| Lateral frontal cortex | 0.14 | 0.04 | 15.86 | <0.001 | Lateral frontal cortex | 0.56 | 0.05 | 54.81 | <0.001 |
| Medial frontal cortex | 0.20 | 0.05 | 17.41 | <0.001 | Medial frontal cortex | 0.58 | 0.06 | 44.94 | <0.001 |
| Lateral parietal cortex | 0.25 | 0.05 | 21.64 | <0.001 | Lateral parietal cortex | 0.68 | 0.03 | 97.91 | <0.001 |
| Occipital cortex | 0.37 | 0.06 | 27.52 | <0.001 | Occipital cortex | 0.79 | 0.06 | 63.31 | <0.001 |
| HCP diffusion | | | | | HCP diffusion | | | | |
| Temporal cortex | 0.23 | 0.05 | 21.52 | <0.001 | Temporal cortex | 0.47 | 0.08 | 26.32 | <0.001 |
| Lateral frontal cortex | 0.07 | 0.04 | 8.44 | <0.001 | Lateral frontal cortex | 0.38 | 0.06 | 29.60 | <0.001 |
| Medial frontal cortex | 0.12 | 0.05 | 11.34 | <0.001 | Medial frontal cortex | 0.36 | 0.07 | 21.59 | <0.001 |
| Lateral parietal cortex | 0.17 | 0.05 | 14.26 | <0.001 | Lateral parietal cortex | 0.58 | 0.04 | 66.37 | <0.001 |
| Occipital cortex | 0.28 | 0.07 | 18.90 | <0.001 | Occipital cortex | 0.63 | 0.07 | 39.20 | <0.001 |

R coefficients and standard deviations are shown for correlations between original and reconstructed data. T and p-values indicate whether correlations are significantly different from 0.
DCGAN deep convolutional generative adversarial networks, GSP Genomic Superstruct Project, GSR global signal regression, HCP Human Connectome Project.

(top row) and DCGAN-reconstructed (middle row) FC maps sourced from one randomly selected vertex within each of the artificially compromised regions in the same participant. All reconstructed FC maps yielded high similarity to the original ones: $r = 0.96$, $p < 0.001$ for the temporal vertex (also shown in Fig. 2c), $r = 0.84$, $p < 0.001$ for the lateral frontal vertex, $r = 0.89$, $p < 0.001$ for the medial frontal vertex, $r = 0.84$, $p < 0.001$ for the lateral parietal vertex, and $r = 0.88$, $p < 0.001$ for the occipital vertex. We then evaluated the average reconstructive accuracy across all participants. When taking into account all the vertices within the compromised regions, multilevel linear models, corrected for multiple comparisons using the Bonferroni correction, revealed that the cortical FC maps of the reconstructed vertices are highly similar to those of the original vertices in all regions. The average correlation between corresponding FC maps is $r = 0.62 \pm 0.06$ (CI [0.60, 0.65], $t(19) = 47.88$, $p < 0.001$) for the temporal cortex, $r = 0.60 \pm 0.04$ (CI [0.58, 0.62], $t(19) = 65.04$, $p < 0.001$) for the lateral frontal cortex, $r = 0.61 \pm 0.05$ (CI [0.58, 0.63], $t(19) = 51.87$, $p < 0.001$) for the medial frontal cortex, $r = 0.70 \pm 0.03$ (CI [0.68, 0.71], $t(19) = 110.62$, $p < 0.001$) for the parietal cortex, and $r = 0.79 \pm 0.05$ (CI [0.76, 0.81], $t(19) = 68.39$, $p < 0.001$) for the occipital cortex (Table 1).

We also evaluated reconstructive accuracy according to the size of the compromised regions (Supplementary Fig. 1a) by correlating the reconstructed and original FC maps. When there are no compromised regions (0%; Supplementary Fig. 1b), the reconstructive accuracy is very high, $r = 0.85 \pm 0.00$, $p < 0.001$ (CI [0.85, 0.85]). Once the compromised region covers 10% of the cortical surface, the mean reconstructive accuracy drops to $r = 0.51 \pm 0.10$, $p < 0.001$ (CI [0.46, 0.56]). From there, there is a steady decrease in reconstructive accuracy as the mask of compromised regions increases in size ($F(2.62, 23.55) = 93.68$, $p < 0.001$, $\eta_p^2 = 0.91$) (Supplementary Fig. 1b). When the mask size reaches 40% of the cortical surface, the reconstructive accuracy is $r = 0.35 \pm 0.12$, $p < 0.001$ (CI [0.28, 0.41]). It should be noted that a complete signal loss in 40% of the cortical surface may represent an extreme case; nevertheless, the reconstruction still recovers some important characteristics of a given individual's functional connectivity.

We replicated our findings by performing the same time series and FC-based analyses in a data set sporting higher spatial and temporal resolutions: the HCP data set[13]. Again, the data of 80 randomly chosen participants were used to train our DCGAN model, and the data of 20 independent participants were used to test reconstruction. Multilevel linear models revealed significant positive correlations between the original and reconstructed time series in all regions: temporal cortex: $r = 0.31 \pm 0.05$ (CI [0.29, 0.33], $t(19) = 27.60$, $p < 0.001$), lateral frontal cortex:

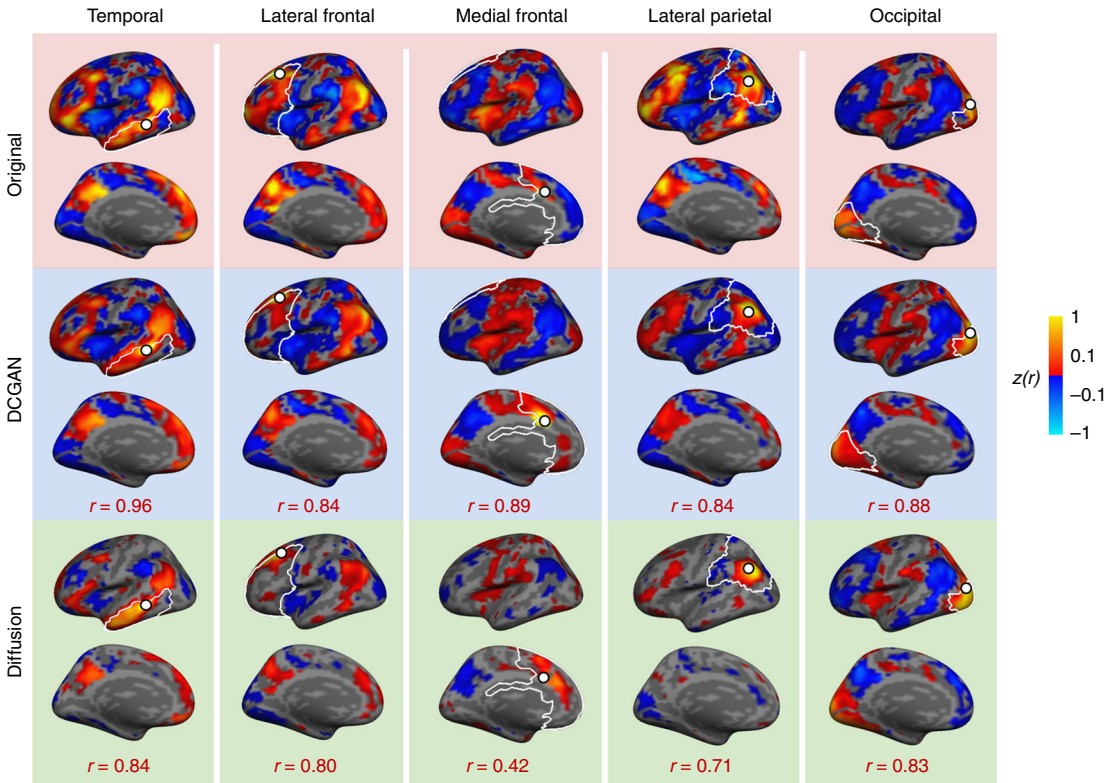

**Fig. 3 The DCGAN-generated functional connectivity maps are highly similar to the original maps, throughout cortex.** The top row shows functional connectivity (FC) maps of seeds from intact BOLD frames in the temporal, lateral frontal, medial frontal, lateral parietal, and occipital cortices. The middle row shows the FC maps of the same seeds extracted from DCGAN-reconstructed BOLD frames. The similarity between the original and reconstructed FC maps is high, with the following correlation coefficients: $r = 0.96$ for the temporal seed, $r = 0.84$ for the lateral frontal seed, $r = 0.89$ for the medial frontal seed, $r = 0.84$ for the lateral parietal seed, and $r = 0.88$ for the occipital seed. The bottom row shows diffusion-reconstructed FC maps. While correlations with original FC maps are also high, they are consistently lower than DCGAN correlations: $r = 0.84$ for the temporal seed, $r = 0.80$ for the lateral frontal seed, $r = 0.42$ for the medial frontal seed, $r = 0.71$ for the lateral parietal seed, and $r = 0.83$ for the occipital seed.

$r = 0.12 \pm 0.04$ (CI [0.11, 0.14], $t(19) = 14.33$, $p < 0.001$), medial frontal cortex: $r = 0.18 \pm 0.05$ (CI [0.16, 0.20], $t(19) = 16.73$, $p < 0.001$), parietal cortex: $r = 0.24 \pm 0.05$ (CI [0.22, 0.27], $t(19) = 19.71$, $p < 0.001$), and occipital cortex: $r = 0.35 \pm 0.07$ (CI [0.32, 0.39], $t(19) = 23.39$, $p < 0.001$) (Table 1). We reconstructed FC maps, and multilevel linear models again revealed significant positive correlations in all regions: temporal cortex: $r = 0.60 \pm 0.07$ (CI [0.57, 0.63], $t(19) = 38.04$, $p < 0.001$), lateral frontal cortex: $r = 0.44 \pm 0.05$ (CI [0.42, 0.47], $t(19) = 39.47$, $p < 0.001$), medial frontal cortex: $r = 0.46 \pm 0.06$ (CI [0.43, 0.49], $t(19) = 32.53$, $p < 0.001$), parietal cortex: $r = 0.62 \pm 0.03$ (CI [0.60, 0.63], $t(19) = 81.92$, $p < 0.001$), and occipital cortex: $r = 0.65 \pm 0.06$ (CI [0.62, 0.68], $t(19) = 46.90$, $p < 0.001$) (Table 1). Comparing the HCP and GSP reconstructions, the GSP-trained model yielded marginally more accurate time series reconstructions when correcting for multiple comparisons (mean difference = 0.02, CI [0.004, 0.04], $t(30.21) = 2.41$, $p = 0.022$, $\eta^2 = 0.16$; significant $p$ threshold = 0.017) and significantly more accurate FC map reconstructions (mean difference = 0.11, CI [0.10, 0.13], $t(38) = 12.88$, $p < 0.001$, $\eta^2 = 0.81$) (Fig. 4a), despite the HCP data set having higher spatial and temporal resolutions. We postulated that the HCP data may have a lower temporal signal-to-noise ratio (tSNR) than the GSP data set, and indeed this was the case ($t(139.43) = 49.31$, bootstrapped $p = 0.001$, $\eta^2 = 0.95$) (Fig. 4b). To counteract this, we temporally smoothed the HCP data by averaging together every 4 frames before retraining and retesting the model. This significantly increased tSNR ($t(99) = -40.68$, $p < 0.001$, $\eta^2 = 0.94$) (Fig. 4b) and yielded more accurate reconstructions than the raw HCP-trained model (time series: mean

difference $= -0.01$, CI $[-0.02, -0.01]$, $t(19) = -4.67$, $p < 0.001$, $\eta^2 = 0.53$; FC maps: mean difference $= -0.09$, CI $[-0.10, -0.09]$, $t(19) = -143.21$, $p < 0.001$; $\eta^2 = 1.00$) (Fig. 4a; see Table 1 for reconstruction details on each cortical region). The tSNR of the temporally smoothed HCP data remained lower than the GSP's ($t(198) = 23.26$, bootstrapped $p = 0.001$, $\eta^2 = 0.73$) (Fig. 4b), however, the time series reconstructions are similar in accuracy to the GSP's (mean difference = 0.01, CI $[-0.008, 0.03]$, $t(32.12) = 1.16$, $p = 0.26$, $\eta^2 = 0.04$), while the reconstructed FC maps are marginally less accurate after Bonferroni correction (mean difference = 0.02, CI [0.003, 0.04], $t(38) = 2.43$, $p = 0.020$, $\eta^2 = 0.13$; significant $p$ threshold = 0.017) (Fig. 4a). These findings indicate that tSNR has an important effect on machine learning reconstructive accuracy.

To assess the power of DCGAN, we compared its performance to a simpler diffusion-based method for filling in compromised cortical regions (see "Methods" section). The diffusion model was able to reconstruct both time series and FC maps (Table 1). As predicted, its reconstructions are less accurate than the DCGAN model's (time series: mean difference = 0.07, CI [0.06, 0.09], $t(19) = 9.15$, $p < 0.001$, $\eta^2 = 0.82$; FC maps: mean difference = 0.18, CI [0.16, 0.19], $t(19) = 27.62$, $p < 0.001$, $\eta^2 = 0.98$) (Fig. 5). As an example, in Fig. 3 we show FC maps generated from (i) original (top row), (ii) DCGAN-reconstructed (middle row), and (iii) diffusion-reconstructed (bottom row) vertices in all five cortical areas. We replicated these results in the raw HCP data set (reconstructed time series: mean difference = 0.07, CI [0.065, 0.067], $t(19) = 176.20$, $p < 0.001$, $\eta^2 = 1.00$; reconstructed FC maps: mean difference = 0.06, CI [0.059, 0.064], $t(19) = 53.50$,

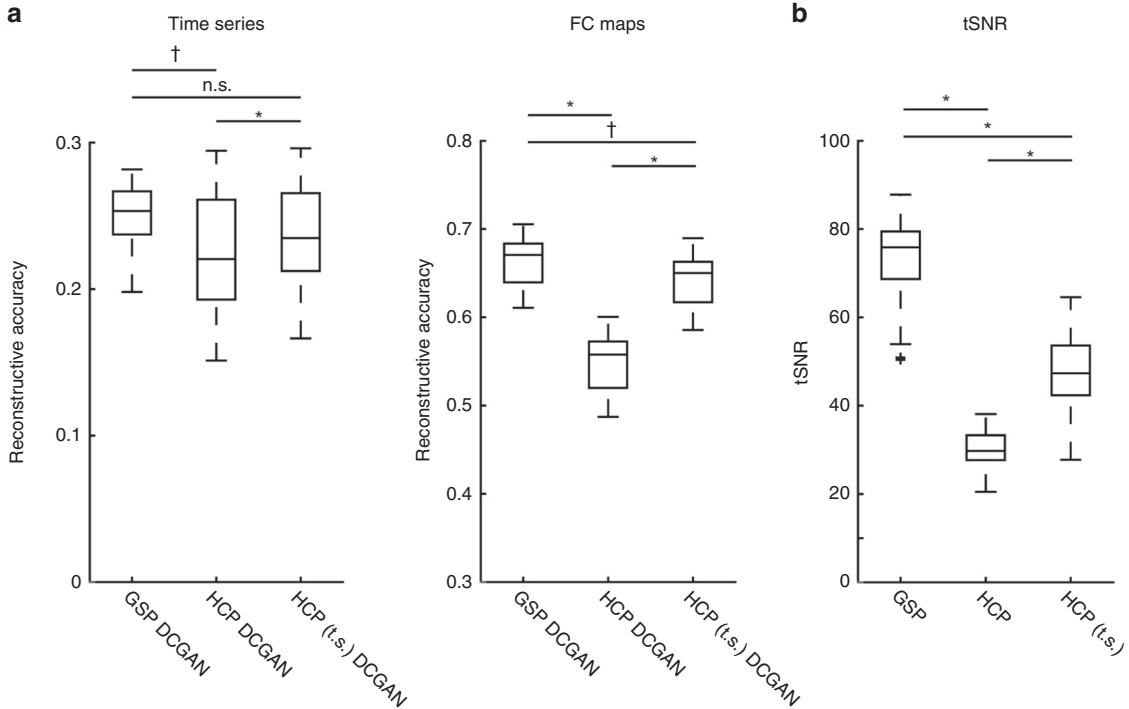

**Fig. 4 DCGAN successfully reconstructs images across data sets, and the temporal signal-to-noise ratio (tSNR) of the training data modulates reconstructive accuracy.** Box-and-whisker plots are shown, with the center line indicating the median, box limits indicating upper and lower quartiles, whiskers indicating 1.5 times the interquartile range, and plus signs indicating outliers. **a** Brain Genomics Superstruct Project (GSP)'s reconstructive accuracy is generally greater than the Human Connectome Projects (HCP)'s (time series: ($t(30.21) = 2.41$, $p = 0.022$) (marginally significant after Bonferroni correction); FC maps: $t(38) = 12.88$, $p < 0.001$). Its reconstructive accuracy for time series is non-significantly different from the temporally smoothed (t.s.) HCP's ($t(32.12) = 1.16$, $p = 0.26$), and its FC map reconstructive accuracy is marginally greater when correcting for multiple comparisons ($t(38) = 2.43$, $p = 0.020$). Temporally smoothed HCP data yielded more accurate reconstructions than the raw HPC data (time series: $t(19) = -4.67$, $p < 0.001$; FC maps: $t(19) = 143.21$, $p < 0.001$). **b** In accordance with these results, the temporal signal-to-noise ratio (tSNR) is higher in the data sets that yielded more accurate reconstructive accuracy. The tSNR of the GSP data set is greater than both the raw HCP ($t(139.43) = 49.31$, bootstrapped $p = 0.001$), and t.s. HCP ($t(198) = 23.26$, bootstrapped $p = 0.001$) data sets. Temporally smoothing the HCP data significantly increased its tSNR ($t(99) = -40.68$, $p < 0.001$). All statistical tests were two-sided and corrected for multiple comparisons. Source data are provided as a Source Data file. *$p \leq 0.001$. †Marginally significant with Bonferroni-corrected $p$-value. n.s. non-significant.

$p < 0.001$, $\eta^2 = 0.99$) (Fig. 5 and Table 1). This finding suggests that our DCGAN model extrapolates information embedded within nearby as well as distant cortical regions to reproduce patterns of brain activity, while more naive methods can only rely on nearby information.

The resting-state fMRI data used for reconstruction were preprocessed with global signal regression (GSR), which introduces spurious temporal anticorrelations in FC analysis[14]. While progress has been made, there is still no consensus about whether GSR should or should not be included in resting-state data preprocessing[15]. To ensure the robustness of our results, we retrained our DCGAN model using the same data, albeit preprocessed without GSR. This data yielded lower reconstructive accuracy for time series (mean difference = −0.001, CI [−0.003, −0.0004], $t(19) = -2.81$, $p = 0.01$, $\eta^2 = 0.29$) and higher reconstructive accuracy for FC maps (mean difference = 0.003, CI [0.002, 0.005], $t(19) = 4.87$, $p < 0.001$, $\eta^2 = 0.55$) compared to GSR-preprocessed data (Supplementary Fig. 2; see Table 1 for reconstruction details on each cortical region). Importantly, however, the effect was negligible in both cases, as evidenced by the near-zero mean differences in $r$ coefficients: −0.001 (CI [−0.003, −0.0004]) for time series and 0.003 (CI [0.002, 0.005]) for FC maps (Supplementary Fig. 2). This finding indicates that GSR does not affect machine learning reconstructive accuracy in any meaningful way and that the information learned by the

DCGAN model is stable. The remaining analyses were conducted on GSR-preprocessed GSP data.

**Reconstructed BOLD signals are individual-specific.** We investigated whether the reconstructed BOLD signals reflect general trends in BOLD activity or whether they capture patterns of activity that are specific to the individual participant. To test this, for each vertex within a compromised region, we calculated the correlation between each individual's reconstructed FC map and (i) their original intact FC map; (ii) the group-averaged FC map from the training data set; and (iii) the FC map of each participant's most similar individual (MSI; see "Methods" section), i.e., the individual in the training data set that most resembles their functional connectivity patterns. In Fig. 6a, we show examples of FC maps and correlations for the same two vertices as in the BOLD time series analyses above. The reconstructed FC maps show the highest correlations with the original FC maps. In Fig. 6b, we make the same comparisons but this time across all vertices within the five cortical masks combined using a repeated-measures ANOVA. Reconstructed and original FC maps share many features and exhibit an average correlation of $r = 0.66 \pm 0.03$ (CI [0.65, 0.68]), which is significantly different from 0 ($t(19) = 116.77$, $p < 0.001$). The average correlation between the reconstructed FC maps and the training group-averaged FC maps is $r = 0.44 \pm 0.02$ (CI [0.44, 0.46]) and is also significantly different from 0 ($t(19) = 85.33$, $p < 0.001$).

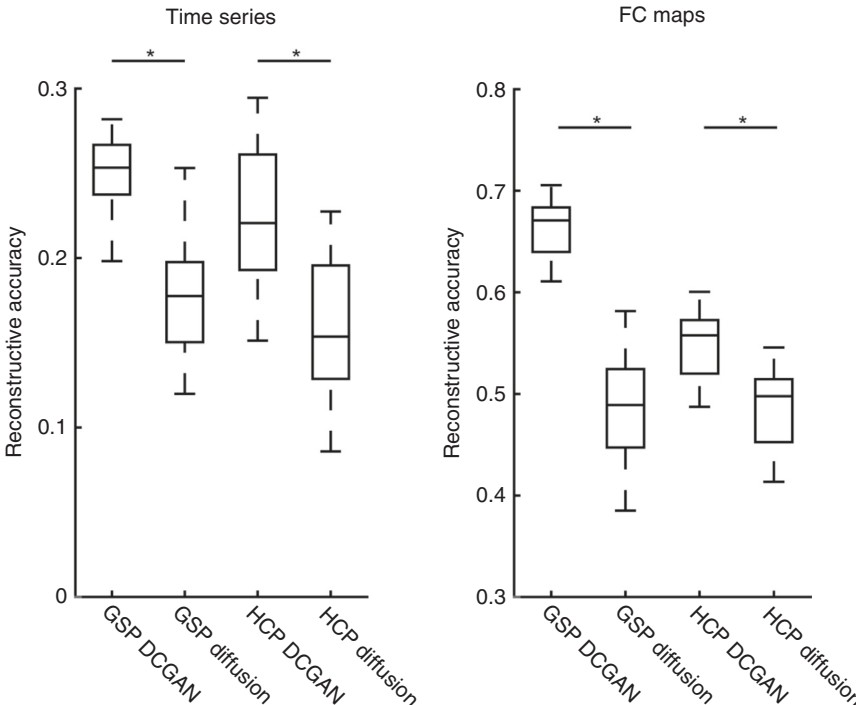

**Fig. 5 The DCGAN model outperforms a diffusion-based model.** Box-and-whisker plots are shown, with the center line indicating the median, box limits indicating upper and lower quartiles, whiskers indicating 1.5 times the interquartile range, and plus signs indicating outliers. DCGAN-reconstructed time series is more similar to the original ones than the diffusion-reconstructed time series, in both the GSP data set ($t(19) = 9.15$, $p < 0.001$) and the HCP data set ($t(19) = 176.20$, $p < 0.001$). The DCGAN model also outperforms the diffusion model when reconstructing functional connectivity (FC) maps, in both the GSP ($t(19) = 27.62$, $p < 0.001$) and HCP ($t(19) = 53.50$, $p < 0.001$) data sets. All statistical tests were two-sided and corrected for multiple comparisons. Source data are provided as a Source Data file. $*p \leq 0.001$.

Finally, we compared the reconstructed FC maps of each individual in the test data set to the FC maps of their MSI. The average correlation between reconstructed and MSI FC maps is $r = 0.55 \pm 0.02$ (CI [0.54, 0.56]), which is significantly different from 0 ($t(19) = 116.86$, $p < 0.001$). We statistically compared these correlations and found them to be significantly different from one another ($F(2,38) = 927.69$, $p < 0.001$, $\eta_p^2 = 0.98$). Post-hoc tests (bootstrapped paired $t$-tests) revealed that the correlation between reconstructed and original FC maps is greater than all other correlations (reconstructed vs. training group-averaged maps: mean difference = 0.22, CI [0.21, 0.23], $t(19) = 34.53$, bootstrapped $p = 0.001$, $\eta^2 = 0.98$; reconstructed vs. MSI maps: mean difference = 0.11, CI [0.10, 0.12], $t(19) = 20.70$, bootstrapped $p = 0.001$, $\eta^2 = 0.96$). The fact that the correlation between reconstructed and original FC maps is higher than the correlation between reconstructed and training group FC maps indicates that during the training process, the generator did not simply learn general trends in BOLD activity but was able to infer the co-activating patterns from the individual-specific BOLD frames used as input in the test phase. Critically, the reconstructed BOLD FC maps are more representative of each individual's own specific patterns of functional connectivity than of any other individual in the training data set, indicating that the DCGAN model is able to capture individual-specific information about functional brain organization.

**Signals are successfully reconstructed in a clinical sample.** We tested the DCGAN model in 12 patients with Parkinson's disease (PD) whose BOLD signals were interfered with by metal implants. Intracortical electrodes were implanted in these patients for DBS treatment[16,17]. Wires outside the skull connecting the

simulator to the implanted electrodes strongly interfere with the acquisition of the BOLD signal during post-surgical fMRI studies, resulting in a signal loss in temporal, parietal, and occipital regions (see Fig. 7 for examples in two representative patients) and in abnormal measurements of functional connectivity. Extensive resting-state fMRI data were acquired both before and after electrode implantation surgery in these 12 patients. We first identified the compromised region where vertices exhibited a sharp contrast in signal amplitude before and after the electrode implantation surgery. The compromised region covered on average 8.36% of the cortical surface.

Once the DCGAN model reconstructed the BOLD signals in the compromised regions, we investigated whether there was any residual loss within these regions. We compared the normalized amplitudes of the pre-operative BOLD signal with that of (i) the post-operative compromised BOLD signal (Supplementary Fig. 3a, left), and (ii) the post-operative reconstructed BOLD signal (Supplementary Fig. 3a, right). While the post-operative signal demonstrated substantial attenuation within the DBS-compromised cortical regions, the reconstruction displayed no residual signal loss. Using the pre-operative average signal amplitudes as a reference, we calculated the normalized BOLD amplitudes (%) of the masked-out vertices in the post-operative images and reconstructed images. A multilevel linear model shows that the BOLD amplitudes are significantly different under pre-operative, post-operative, and reconstructed conditions ($F(2,1555.89) = 9695.62$, $p < 0.001$). Examination of the parameter estimates revealed that the post-operative BOLD amplitudes are substantially and significantly attenuated in comparison to the pre-operative BOLD amplitudes (mean difference = $-32.54$, CI [$-33.08$, $-32.01$], $t(1554.80) = -119.98$, $p < 0.001$, $\eta^2 = 0.90$). The reconstructed BOLD signals are not significantly different

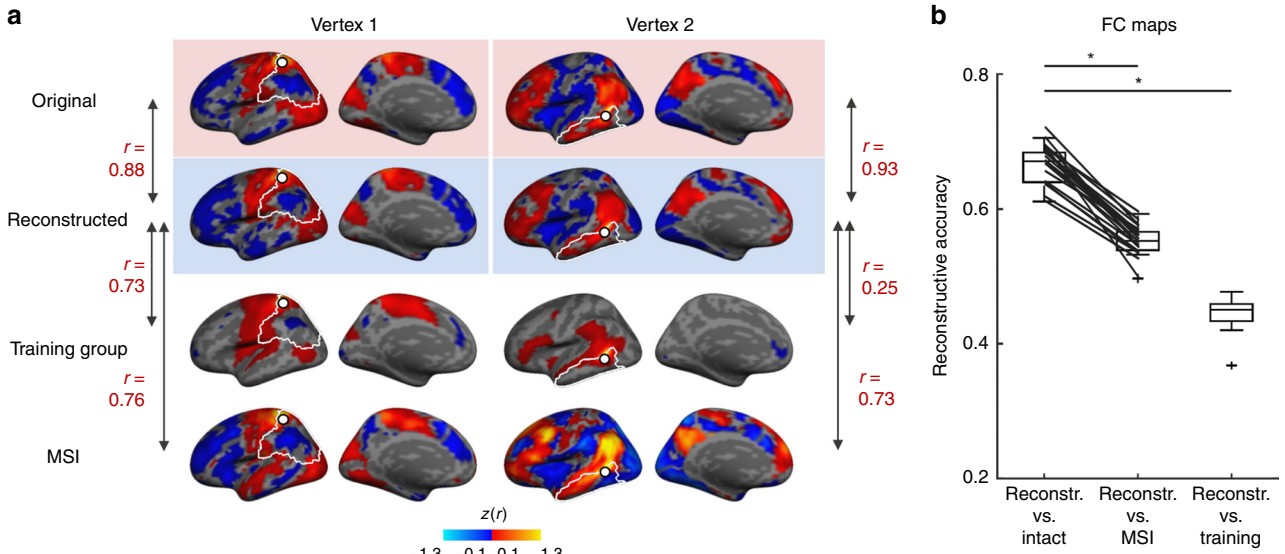

**Fig. 6 DCGAN-reconstructed functional connectivity maps capture individual-specific information. a** For each participant in the test data set, we generated functional connectivity (FC) maps from all vertices within each of the compromised regions. Here, we show FC maps calculated from two randomly selected seeds on the cortical surface, one in the motor region of the lateral parietal cortex and one in the lateral temporal cortex. We generated FC maps from intact BOLD frames (first row), reconstructed BOLD frames (second row), group-averaged functional connectivity data (third row), and data from each participant's most similar individual (MSI) from the training data set. To identify each test participant's MSI, we correlated that participant's FC vectors inside the compromised regions with the corresponding vectors in each of the training data set individuals. The MSI was the individual displaying the highest average correlation. **b** If reconstructed BOLD frames capture specific details about an individual's functional brain organization, we would expect the reconstructed FC maps to be more similar to the intact FC maps than to any other FC map. Box-and-whisker plots are shown, depicting the reconstructive accuracy for reconstructed FC maps relative to the intact, MSI, and training group FC maps. The center line indicates the median, box limits indicate upper and lower quartiles, whiskers indicate 1.5 times the interquartile range, and plus signs indicate outliers. As expected, we found that the correlation between participants' reconstructed and intact FC maps, averaged over all vertices within the five cortical masks ($r = 0.66 \pm 0.03$), was significantly higher than the correlation between reconstructed and MSI FC maps ($r = 0.55 \pm 0.02$; $t(19) = 20.70$, bootstrapped $p = 0.001$) and also significantly higher than the correlation between reconstructed and training group-averaged FC maps ($r = 0.44 \pm 0.02$; $t(19) = 34.53$, bootstrapped $p = 0.001$). The diagonal lines between the first two boxes join data points from the same participants. All statistical tests were two-sided and corrected for multiple comparisons. Source data are provided as a Source Data file. *$p \leq 0.001$.

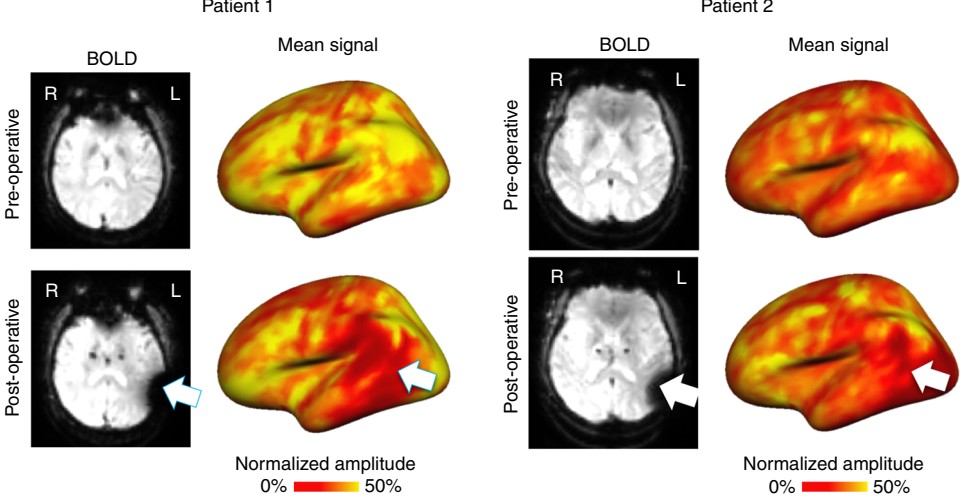

**Fig. 7 Implanted electrode interference severely reduces BOLD signal amplitude.** Pre- and post-operative BOLD frames are shown in the left panels for two representative patients with Parkinson's disease. Patients had electrodes implanted in the subthalamic nucleus, resulting in interference around left temporal, parietal, and occipital regions. The pre- and post-operative normalized and averaged BOLD signals are shown on inflated cortical surfaces in the right panels. Following surgery, the BOLD signal amplitude is severely affected in regions close to the electrode wires (temporal and parietal regions for patient 1; temporal, parietal, and occipital regions for patient 2).

from the pre-operative BOLD signals (mean difference $= -0.33$, CI $[-0.20, 0.86]$, $t(1554.44) = 1.21$, $p = 0.23$, $\eta^2 = 0.0009$). The average pre-operative, post-operative, and reconstructed normalized BOLD amplitudes are shown in Supplementary Fig. 3b.

Next, we sought to evaluate the reconstructive accuracy using functional connectivity analyses, similar to those conducted in the healthy cohorts above. We did not consider the BOLD time series here as the pre- and post-operative fMRI scans were obtained at

different time points. However, functional brain organization, as assessed with functional connectivity, is assumed to be relatively stable over time[18]. Although the implantation surgery may cause microlesion effects and lead to changes in brain circuits involving the stimulation target (i.e., the subthalamic nucleus), the surgery is unlikely to change functional connectivity in the area of signal loss, which is relatively far from the location of the stimulator and the motor network being modulated. As a proof of concept, we investigated the functional connectivity of the right temporoparietal region in the same two representative patients (note that signal loss was observed only in the left hemisphere). We found that the post-operative FC maps in the right hemisphere are highly and significantly correlated with the pre-operative FC maps, $r = 0.75$, standard error $= 0.02$ (CI [0.71, 0.79], $t(10.98) = 42.96$, $p < 0.001$, $\eta^2 = 0.99$). However, for the compromised region in the left hemisphere, we found that the post-operative FC maps are only weakly positively correlated with the pre-operative FC maps, $r = 0.37$, standard error $= 0.03$ (CI [0.30, 0.44], $t(11.02) = 11.24$, $p < 0.001$, $\eta^2 = 0.92$). As an example, we show cortical FC maps using a seed placed in two representative patients' compromised regions (Supplementary Fig. 4). Unlike the weak and disorganized FC maps obtained from patients' compromised left temporoparietal region, the FC maps generated from seeds in the right temporoparietal region show high similarity to their pre-operative FC maps (right hemisphere seeds across the two patients in Supplementary Fig. 4: $r = 0.83 \pm 0.03$, $p < 0.001$; left hemisphere seeds across both patients: $r = 0.38 \pm 0.13$, $p < 0.001$) (Supplementary Fig. 4).

Having shown that FC maps are relatively stable following electrode implantation, we next assessed the reconstructive accuracy of our DCGAN model for the patients' compromised region in the left hemisphere. A multilevel linear model that took into account all the vertices inside the compromised regions showed that the reconstructed post-operative FC maps are moderately positively correlated with the pre-operative FC maps, $r = 0.56$, standard error $= 0.02$ (CI [0.52, 0.60]), and the correlation is significantly different from 0 ($t(10.83) = 30.50$, $p < 0.001$, $\eta^2 = 0.99$). As mentioned above, the post-operative FC maps were also positively correlated with the pre-operative FC maps, but this correlation was weak ($r = 0.37$, standard error $= 0.03$; CI [0.30, 0.44], $t(11.02) = 11.24$, $p < 0.001$, $\eta^2 = 0.92$). Another multilevel linear model showed that the correlations between reconstructed and pre-operative FC maps are significantly higher than the correlations between post-operative and pre-operative FC maps (mean difference $= 0.18$, CI [0.15, 0.20], $t(667.35) = 13.63$, $p < 0.001$, $\eta^2 = 0.22$). As an example, we show cortical FC maps using a seed placed in two representative patients' compromised regions (Fig. 8, same patients and seeds as in Fig. 7 and Supplementary Figs. 3 and 4). As expected, the FC maps obtained from the patients' compromised post-operative BOLD images do not capture the connectivity patterns observed in the pre-surgical data. However, the FC maps generated from the reconstructed BOLD images show high similarity to the pre-operative FC maps (reconstructed vs. pre-operative across the two patients in Fig. 8: $r = 0.61 \pm 0.01$; post-operative vs. pre-operative across both patients: $r = 0.38 \pm 0.13$). These results indicate that the BOLD signals reconstructed in the compromised regions are representative of the patients' intact functional connectivity patterns.

## Discussion

The current study aimed to reconstruct fMRI BOLD signal inside cortical regions that suffered a signal loss due to various artifacts. We used deep convoluted generative adversarial networks (DCGAN), a recent advance in machine learning algorithms, to leverage functional information embedded within BOLD frames and reconstruct the signal in compromised regions, frame by frame. We reconstructed BOLD signals in three cohorts: healthy young adults (GSP and HCP data sets) with artificially compromised cortical regions and patients whose fMRI scans suffered from interference due to metal implants. Our results indicate that such a machine learning technique successfully reconstructs individual-specific BOLD signals and can approximate the functional connectivity patterns observed in the intact or unimpaired state.

The missing BOLD signal was reconstructed frame by frame, following which we modeled the time series for all individual vertices whose signal was compromised. We found the reconstructed time series to be similar to the original time series. This indicates that our model was able to recover dynamic brain activity over time, at the level of single vertices within individual participants. The same was found for the reconstructed and original functional connectivity maps. These findings are intriguing, as the images were reconstructed at each time point independently and single input frames did not carry time-varying information. The generator was thus able to learn information beyond what was presented at face value during the test phase, and modeled accurate functional interactions between brain regions and the changing dynamics of the BOLD signal through time. The DCGAN model outperformed a more naive diffusion-based reconstruction method, indicating that machine learning is able to extrapolate information embedded within the whole BOLD image, while simpler filling-in methods are restricted to using nearby information, thereby limiting their ability to capture principles of brain organization. Additional control analyses revealed that global signal regression (GSR), a preprocessing step that amplifies anticorrelations in the brain through its mathematical mandate[14,15], does not meaningfully impact the learning of these principles. Of note, the temporal signal-to-noise ratio (SNR) of the training data is an important factor that modulates reconstructive accuracy.

The reconstruction of the compromised signals is based on learning the functional relationships between different brain regions in intact BOLD frames from a large data set. The trained model deduces possible signal distributions in the compromised region using the remaining intact BOLD amplitude patterns in the individual. Thus, the generator learns functional activity patterns that are specific to each individual and builds a high-dimensional space that is sensitive to variations across individuals. Indeed, we show that the reconstructed BOLD signals capture individual differences in patterns of activity. A given individual's reconstructed FC maps were more similar to their original FC maps than to the training group-averaged maps or to the ones belonging to the most similar individual from the training data set. Therefore, the deep machine learning model may be useful in recovering lost signal in clinical settings, where the focus is on the individual.

On that note, our method has several potential clinical applications. Revealing individual-based functional activity is critical not only for understanding the functional network organization of the human brain[19] but also for personalized medicine, such as when precise cortical mapping is required for neurosurgery or neuromodulation[20,21]. Our individual-specific machine learning method can, as we have shown, reconstruct the BOLD signal that was lost due to intracortical electrode interference. We showed that the model generates FC maps with high reconstructive accuracy, as they exhibit high similarity to maps derived from pre-surgical images. Thus, machine learning-based reconstruction can impact the investigation of various disorders where intracortical electrodes are used for diagnosis or therapy, such as in epilepsy, Parkinson's disease, depression, obsessive-compulsive

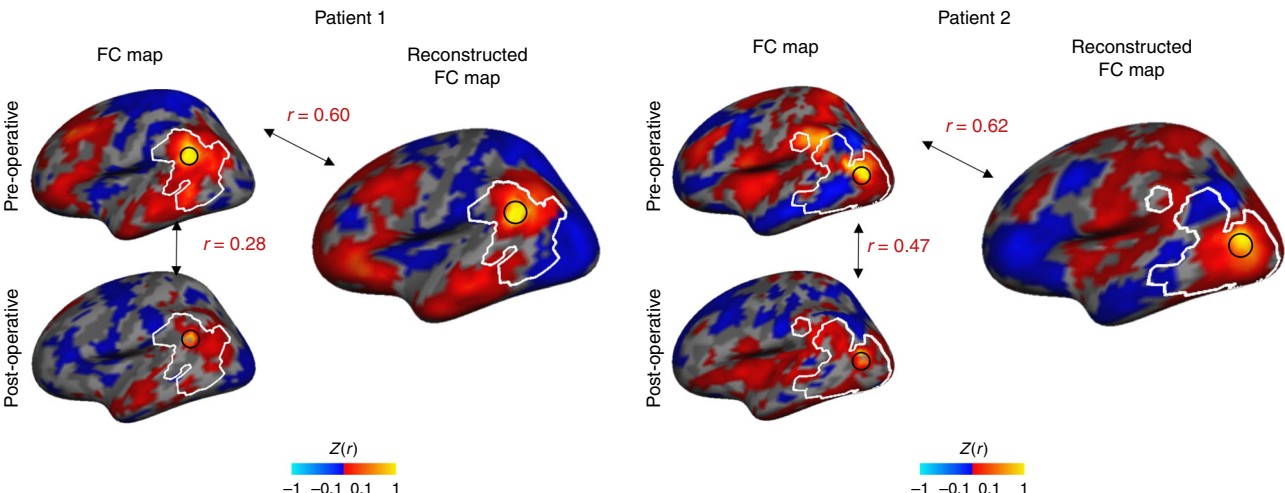

**Fig. 8 The DCGAN model can reconstruct BOLD signals compromised by the implantation of a deep-brain stimulator in patients with Parkinson's disease.** We generated the FC map of a given vertex (black circle) before and after surgery in two representative patients (left panels). Delineated in white is the region that is affected by the implanted device, defined by identifying the vertices which showed a stark decrease in BOLD amplitude after the implantation surgery. The post-operative FC maps are substantially different from the pre-operative FC maps due to the intracortical electrodes, with a relatively low correlation of $r = 0.28$ in patient 1 and $r = 0.47$ in patient 2. We also generated FC maps for the same vertices, this time using the reconstructed BOLD frames (right panels). The generated FC maps are highly similar to the intact pre-operative FC maps, as indicated by a correlation of $r = 0.60$ for patient 1 and $r = 0.62$ for patient 2. Thus, the DCGAN model is able to reconstruct patients' functional brain organizations.

disorder, among others. This method could also help mitigate registration problems in fMRI that occur with data from patients with brain lesions due to stroke or tumor resections, by filling in the compromised regions prior to registration. It could also fill in regions that are routinely cut off during acquisition, such as the top of the brain or the lower portion of the cerebellum. Another application would be to reconstruct whole frames in patient fMRI data in which many frames were scrubbed due to excessive head motion, a common pitfall in clinical studies. Finally, in both clinical and non-clinical settings, machine learning could be used to reconstruct the BOLD signal in regions that are susceptible to signal loss and geometric distortion, such as the orbitofrontal cortex and temporal cortex. However, this could only be done once the model can be trained on uncompromised images, thus appropriate data acquisition methods that can counteract these susceptibility artifacts will first have to be developed.

We observed that the reconstructive accuracy differed from region to region. Reconstructive accuracy may be affected by a number of factors. One factor is the size of the compromised region, as smaller regions yield better reconstructions. A second factor is the shape of the region, as closer proximity to uncompromised vertices is likely to result in better reconstruction. A third factor is whether the affected region has important large-scale network connectivity, which would increase accuracy as fMRI activity outside of the compromised region would bear information relevant to the activity within the compromised region. Additionally, the distribution of learned functional activation patterns is constrained by the training data set. More efforts are necessary to evaluate the reconstructive accuracy of the model when the compromised BOLDs are not acquired using the same MRI and scanning parameters as the training data. Another limitation of our study is that the DCGAN model dealt with 2D images. Currently, it is not able to reconstruct whole-brain 3D images as the computational power required is too high. We hope that technological advances will soon enable this type of modeling. In the meantime, it may be possible to reconstruct signal in small 3D volumes.

Besides 3D modeling, we propose one area of future study to improve the current machine learning model, which would be to

consider the causal interactions across time frames. Currently, each of the BOLD frames is used separately to supervise the learning process of the adversarial networks. In this way, the connections among cortical areas, which are not included in single BOLD frames, are difficult to detect. Feeding DCGAN a combination of BOLD frames may improve their modeling power.

In summary, we harnessed the learning power of deep convolutional neural networks to generate BOLD signals in regions that experienced signal loss. We have replicated our findings in multiple data sets and have shown that it is possible to reconstruct lost BOLD signal in healthy individuals as well as in a clinical sample with compromised fMRI. In all cases, the reconstructed signals closely resemble the uncompromised signals. Notably, the reconstructed signals are coherent with each individual's functional brain organization. Such a method could benefit personalized clinical and non-clinical studies where brain regions suffer signal dropout, distortion, or deformation.

## Methods

**Participants.** We used the resting-state fMRI data of 100 healthy young adult participants randomly chosen from the Brain Genomics Superstruct Project (GSP)[10] data set (50 women, 50 men; mean age: 22.0 ± 3.2 years), as well as the "100 Unrelated Subjects" data set[22] that was taken from the large publicly available Human Connectome Project[13] database (54 women, 46 men; age range 22–36). We also analyzed the MRI data of 12 patients with Parkinson's disease (PD; 5 women, 7 men; mean age = 55.3 ± 7.5) from a previous clinical trial (https://clinicaltrials.gov/ct2/show/NCT02937727), who had intracortical electrodes implanted for deep-brain stimulation (DBS) treatment. The patients underwent stereotactic implantation of quadripolar DBS electrodes (PINS Medical, Model L301C) in the subthalamic nucleus. Microelectrode recording and stimulation guided the electrode implantation, and the electrodes were connected to extension leads (PINS Medical, Model E202C), which were themselves connected to the implanted pulse generator (PINS Medical, Model G106R). All participants provided written informed consent in accordance with guidelines set by the Institutional Review Boards of Harvard University, Partners Healthcare, or Beijing Tiantan Hospital of Capital Medical University.

**MRI data acquisition.** GSP data set. Each healthy young participant from the GPS data set underwent one structural scan and two resting-state fMRI scans (6 min and 12 s per scan). Data were collected on matched 3 T Tim Trio scanners (Siemens, Erlangen, Germany) using a 12-channel phased-array head coil. Structural data included a high-resolution multi-echo T1-weighted magnetization-prepared gradient-echo image (TR = 2200 ms, TE = 1.54 ms for image 1 to 7.01 ms for

image 4, TI = 1100 ms, flip angle = 7°, voxel size: 1.2 × 1.2 × 1.2 mm, FOV = 230, slices = 720). Resting-state fMRI images were acquired using the gradient-echo echo-planar imaging (EPI) pulse sequence (TR = 3000 ms, TE = 30 ms, flip angle = 85°, voxel size: 3 × 3 × 3 mm voxels, FOV = 216, slices = 47 slices collected with the interleaved acquisition with no gap between slices). Whole-brain coverage including the entire cerebellum was achieved with slices aligned to the anterior commissure-posterior commissure plane using an automated alignment procedure, ensuring consistency across participants[23]. Participants were instructed to stay awake, keep their eyes open, and minimize head movement; no other task instruction was provided.

HPC data set. HCP participants underwent structural MRI scans (20 min) and two resting-state fMRI scans (30 min each) on a 3 T Siemens Skyra MRI scanner equipped with a 32-channel head coil. Structural images were acquired using a 3D MPRAGE T1-weighted sequence (TR = 2400 ms, TE = 2.14 ms, TI = 1000 ms, flip angle = 8°; voxel size: 0.7 × 0.7 × 0.7 mm, FOV = 224 mm, matrix = 320, 256 sagittal slices in a single slab). Functional images were acquired using a multiplexed EPI pulse sequence (TR = 720 ms, TE = 33.1 ms, flip angle = 52°, voxel size: 2 × 2 × 2 mm, FOV = 208 × 180 mm, 72 slices, multiband factor = 8, echo spacing = 0.58 ms, bandwidth = 2290 Hz/px). To match the duration of the GSP data, we truncated the HCP resting-state data to 8 min.

Clinical data set. Each of the 12 patients with Parkinson's disease underwent four resting-state fMRI scans (6 min and 8 s per scan) at two time points: at baseline before the DBS electrode implantation surgery and one month after. The patients were instructed to stay awake and keep their eyes open. The deep-brain stimulator was turned off during post-surgical fMRI scanning. The specific absorption rate-estimated values were continuously monitored throughout the scanning sessions. MRI data were collected on a Philips Achieva 3.0 Tesla TX whole-body MR scanner using a 32-channel receive-only head coil. Structural images were acquired using a sagittal magnetization-prepared rapid gradient-echo T1-weighted sequence (TR = 7.6 ms, TE = 3.7 ms, TI = 1000 ms, flip angle = 8°, voxel size = 1 × 1 × 1 mm, FOV = 256, slices = 180). Functional data were collected using an echo-planar imaging sequence (TR = 2000 ms, TE = 30 ms, flip angle = 90°, voxel size = 2.875 × 2.875 × 4 mm, FOV = 230, slices = 37).

**Data processing.** Structural data were processed using FreeSurfer version 4.5.0. Surface mesh representations of the cortex from each individual participant's structural images were reconstructed and registered to a common spherical coordinate system[24]. The structural and functional images were aligned using boundary-based registration using the FsFast software package (http://surfer.nmr. mgh.harvard.edu/fswiki/FsFast)[25]. The preprocessed resting-state fMRI data were then aligned to the common spherical coordinate system via sampling from the middle of the cortical ribbon in a single interpolation step[26]. We registered each individual's fMRI data to the FreeSurfer template which consisted of 40,962 vertices in each hemisphere. A 6-mm full-width half-maximum (FWHM) smoothing kernel was applied to the fMRI data in the surface space. The smoothed data were downsampled to a mesh of 2562 vertices in each hemisphere using the mri_-surf2surf function in FreeSurfer.

Resting-state fMRI data were processed using the following procedures: (i) slice timing correction (SPM2; Wellcome Department of Cognitive Neurology, London, UK)[27]; (ii) rigid body correction for head motion in the FSL package[28,29]; (iii) normalization of global mean signal intensity across runs; and (iv) bandpass temporal filtering (0.01–0.08 Hz), head-motion regression, whole-brain global signal regression (GSR), and ventricular and white-matter signal regression in a single step. To test the effects of GSR, we also preprocessed the data without GSR. After preprocessing, each participant's resting-state fMRI data were normalized to [−1,1] by dividing the BOLD amplitude of each vertex by the maximum absolute BOLD value observed in each session using Matlab R2014b. The normalized 2562-vertex mesh of the BOLD frames was then flattened to 2-dimensional maps using the tksurfer and mris_flatten functions in FreeSurfer[24].

**Machine learning.** We used DCGAN to reconstruct lost or compromised BOLD information. The neural network modeling was conducted in three steps using Python 3.6.1. In the first step, the training phase, two competing models are trained: a generator and a discriminator (Fig. 1a). The generator is trained to encode BOLD information by feeding it intact BOLD frames from the training data set. Using information within these frames, the generator creates new frames, which the discriminator then classifies as being either authentic (real BOLD frame) or artificial (generator-created BOLD frame). Both the generator and discriminator simultaneously continue training with new frames, and through many iterations, each becomes optimized. Training ends when an optimized discriminator classifies the generated frames into one category or the other at chance level. In the second step, we created artificially compromised BOLD frames by removing the BOLD signal within certain predefined regions, using data from the test data set. In the third step, the signal in the compromised region is reconstructed by feeding the compromised BOLD images to the DCGAN generator (Fig. 1b). The generator then produces new complete frames based on these. Using the mask of the compromised region, the region with a reconstructed signal from the newly generated frame replaces the one in the compromised frame to form a complete BOLD frame which includes the original information (BOLD signal outside of the compromised region) and the newly generated information (BOLD signal inside the

reconstructed region). Following this, we evaluated the similarity between the reconstructed BOLD information and the original intact BOLD information. Each of these steps is described in more detail below.

*Step 1*: Training a generative network to encode BOLD information. Eighty participants were randomly selected from the Brain GSP data set[10] to build the training data set, which consisted of 19,200 intact flattened BOLD frames in vertex space. The data from 20 other participants, again selected randomly from the GSP, constituted the test data set, which was independent from the training data set. We used a DCGAN model[2] to create BOLD frames in vertex space based on each individual participant's data. The generator's goal is to create images similar enough to the original images that the discriminator is forced to randomly classify them as authentic or simulated, while the goal of the discriminator is to correctly classify images as either authentic or simulated.

In mathematical terms, the generator (G) samples data x from the true data population $p_{data}$ and produces parameters. It maps these parameters onto a random vector z, which is sampled from latent space Z, and creates artificial images G(z), which are part of the generated distribution $p_g$. When the discriminator (D) detects a difference between the distributions $p_{data}$ and $p_g$, the generator G tweaks its parameters and generates images that are more similar to the authentic images. This process is repeated until the generator produces a generated distribution $p_g$ that so closely matches the true data distribution $p_{data}$ that the discriminator D is unable to detect a difference and classifies the authentic or generated images G(z) randomly.

Convolutional neural networks (CNN) were used to build the generator and discriminator[4]. During the adversarial training process, the generator and discriminator were trained simultaneously. They were optimized using a Nash equilibrium of costs two-player minimax game with value function V(G, D):

$$\min_G \max_D V(D, G) = \mathbb{E}_{x \sim p_{data}(x)}[\log D(x)] + \\ \mathbb{E}_{z \sim p_z(z)}[\log(1 - D(G(z)))]. \tag{1}$$

The input z was a sample taken from 100 dimensional uniformly distributed noise; in each dimension, the value varied from −1 to 1. The generator projected the input z to a small convolutional representation and then converted the representation into a 500 × 500 pixel image through four-layer fractionally strided convolutions[4]. The discriminator estimated the input images through four-layer-strided convolutions and fed the layers into single sigmoid outputs. Rectified Linear Unit activation was used in the generator's layers, except for the output layer, which used the Tanh function[30]. Leaky rectified activation was used in all of the discriminator's layers[31]. A 64-size batch normalization was used in the training procedure for stabilization[32], and the sigmoid cross entropy was calculated to measure the probability difference between two images. We used an Adam optimizer during the optimizing procedure of the generator and discriminator[33], with a learning rate of 0.0002. The generator's parameters were adjusted twice during each iteration to balance the learning speed between the generator and the discriminator.

*Step 2*: Creating the compromised BOLD frames. To allow the DCGAN model to generate BOLD signals in compromised regions, we created frames in which we removed the BOLD signal in predefined regions. We aimed to test the generator on five regions spread across the cortical surface: the lateral frontal cortex, the medial frontal, cortex, the lateral parietal cortex, the lateral temporal cortex, and the occipital cortex. These were delineated using FreeSurfer's Desikan–Killiany atlas[34]. The Desikan–Killiany atlas labels of the artificially compromised regions are: 4, 13, 19, 20, 21, 28 for the lateral frontal cortex (329 eliminated vertices, 12.8% of the cortical surface); 3,15, 27, 29 for the medial frontal cortex (223 eliminated vertices, 8.7% of the cortical surface); 9, 30, 32 for the lateral parietal cortex (429 eliminated vertices, 16.7% of the cortical surface); 10 and 16 for the lateral temporal cortex (140 eliminated vertices, 5.5% of the cortical surface); and 6, 12, 14, 22 for the occipital cortex (188 eliminated vertices, 7.3% of the cortical surface). Masks M were created separately to mask out each of the eliminated regions and leave intact the other parts of the flattened BOLD activation map. After preprocessing, BOLD signals within the masks were artificially set to 0 to create the compromised frames.

We also sought to evaluate the reconstructive accuracy of our model according to the size of the compromised regions. To do this, we created ten sets of incrementally larger masks (Supplementary Fig. 1a). We selected at random 10 vertices among all 2562 cortical surface vertices, located in various cortical regions. Each of these vertices served as the center of its mask set. Each mask set was comprised of six masks, whose coverage went from 10 to 60% of the cortical surface, with incremental steps of 10% (i.e., 10, 20, 30, 40, 50, 60%). The BOLD signal inside these masks was eliminated to create 10 sets of BOLD images with increasingly larger compromised regions.

Finally, for the patients with Parkinson's disease, we created masks that encompassed the regions in which interference was observed due to the implanted DBS electrodes. We first calculated the absolute values of the surface-based BOLD signal for each vertex, which were then averaged and normalized to [0,1] to achieve a normalized BOLD signal strength for each vertex before and after implantation surgery. Then, the pre- and post-operative BOLD amplitude maps were contrasted and the vertices with strongly reduced amplitude post-operatively were extracted to create a mask. The compromised regions comprised 193 vertices (7.5% of the

cortical surface) in one patient and 200 vertices in the other (7.8% of the cortical surface).

*Step 3*: Reconstructing compromised BOLD signals. The trained generator was used to reconstruct the BOLD frames with artificially compromised regions, by generating an image $G(z)$ with maximal similarity to the original BOLD activation $x$ in the cortical regions outside of the mask. The generator's loss function serves to calculate the divergence between the real and simulated BOLD frames, and was defined as:

$$L = \sum (x.*M - G(z).*M)^2. \tag{2}$$

In order to minimize the loss function, an optimized $z$ must be found in the latent space. After random sampling, $z$ was mapped to $G(z)$ by the trained generator. The location of $z$ was rearranged iteratively during the optimization process using the gradient-descent method to find the most similar generated image $G(z)$ with minimal $L$ to $x$. The number of iterations was set to 500, with a 0.000002 learning rate.

In the reconstruction step, we used 4800 BOLD frames from the test data set of 20 participants. The reconstructed BOLD frames were built from the masked-out flattened images. To do this, we first had to determine the spatial location of each vertex on a flattened BOLD amplitude map. Each vertex was generated and subsequently projected onto a flattened cortical surface map. The coordinates with maximum value in the flattened map were regarded as the location of the generated vertex. Using this method, we determined the correspondence between each of the vertices and their spatial location on the flattened map. This allowed us to then take the flattened map and to project it back into vertices on a BOLD frame. For each participant, the simulated BOLD frames were assembled temporally to reconstruct the BOLD time series.

**Diffusion model**. We compared our DCGAN model to a more naive reconstruction model based on diffusion, whereby each compromised vertex was assigned a BOLD value based on the average of the BOLD signal in adjacent vertices. The process was started at the perimeter of the compromised region so that those vertices could be filled in using the adjacent uncompromised vertices. Then, reconstruction moved inwards, with new vertices being assigned a BOLD value based on the adjacent vertices that were reconstructed in the previous step. This process was reiterated until all compromised vertices were assigned a value.

**Temporal signal-to-noise ratio**. Temporal signal-to-noise ratio (tSNR) was measured by dividing 1 by the standard deviation of the BOLD signal as described in previous studies[35,36] and then averaged across all voxels within the brain, across frames, and across participants.

**Statistical analysis**. The reconstructive accuracy of the DCGAN model was evaluated by calculating correlations between original and reconstructed BOLD signal time series and functional connectivity. The similarity between the original and reconstructed BOLD time series was only investigated in the first part of the study with healthy adults. The reconstructed signal was compared to the original signal. For the patient study, time series obtained at different time points cannot be directly compared, for this reason, we did not compare the post-operative reconstructed time series to the pre-operative time series in the patients with PD. As the brain's functional connectivity (FC) patterns are relatively stable through time, we compared the FC maps generated from various seeds inside the compromised regions, in both the healthy adult data sets and the PD data set. To show that functional connectivity is relatively stable and unaffected by electrode implantation surgery, we also generated FC maps using seeds in uncompromised regions.

For the time series comparisons at any given vertex, the similarity between reconstructed and original signals was quantified by calculating the Pearson correlation between the two time series. The correlation values within a given masked region were then averaged across participants to represent the similarity between the reconstructed and original BOLD signals in that region.

The cortical FC of the original and reconstructed BOLD signals was also compared. For the original BOLD images, an FC map was created by calculating the $z$ value of the correlation between the BOLD signal at a given vertex and the BOLD signals at all the vertices in the two cerebral hemispheres. For the reconstructed BOLD images, the compromised regions were filled in with the reconstructed BOLD signal to calculate the FC of the whole brain. The vectors storing the FC of the original and reconstructed BOLD signals for a given vertex were then correlated to determine their similarity. For each of the five cortical masks, the statistical significance for the time series and FC map correlations was assessed using multilevel linear models with vertices nested within participants. Differences between various models or data sets were statistically assessed using independent (e.g., GSP vs. HCP) and paired (e.g., raw HCP vs. temporally smoothed HCP, DCGAN vs. diffusion) samples $t$-tests. Whenever the assumption of normality was violated, bootstrapped $p$-values were calculated. Multiples comparisons were corrected for using the Bonferroni correction.

To assess whether the reconstructed BOLD signals were representative of individuals' own BOLD signals or whether they simply reflected general trends in BOLD activity learned from the training data set, we calculated the correlation between the reconstructed BOLD FC map from the test data set and the group-averaged FC map from the training data set in corresponding vertices. The correlations were calculated using all vertices within the five cortical masks combined.

We also compared the reconstructed FC map of each participant in the test data set to the intact FC map of the individual in the training data set that most resembles them, called the most similar individual (MSI). To determine each participant's MSI, we correlated a given individual's functional connectivity vectors for the vertices inside all five cortical masks combined with the vectors of corresponding vertices in each of the 80 training data set participants. The training data set individual showing the highest similarity (highest correlation) to a given participant from the test data set was identified as that participant's MSI. The statistical significance of each of these average correlations (reconstructed vs. original, reconstructed vs. training, reconstructed vs. MSI) was assessed by examining parameter estimates generated from a repeated-measures analysis of variance (ANOVA), which also served to statistically compare whether these three average correlations were different from one another. Finally, we used paired $t$-tests as post-hoc tests to determine which pairs of correlations demonstrated a significant difference. We used a repeated measures ANOVA to determine whether the size of the compromised region significantly affected the reconstructive accuracy of the generated frames across 10 cortical masks.

Statistical tests were performed using SPSS Statistics 20.0 (IBM, NY). All statistical tests were two-sided, and 95% CI are reported. Effect sizes are reported for group-level analyses: $\eta_p^2$ for F-tests, $\eta_p^2$ for $t$-tests, and $r$ coefficients for correlations. Means are presented along with their standard deviations (mean ± s. d.) in the results section, except when indicated otherwise.

## Data availability

The GSP data set is available at http://neuroinformatics.harvard.edu/gsp/. The HCP data set is available at https://www.humanconnectome.org/study/hcp-young-adult/data-releases. The DBS data set is available from the corresponding authors upon reasonable request. Source data are provided with this paper.

## Code availability

The code used in this article is available at http://nmr.mgh.harvard.edu/bid/DownLoad.html.

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

## Acknowledgements

This work is supported by National Natural Science Foundation of China grants 81790650 (H.L.), 81790652 (H.L.), 81527901 (L.L.), and 81720108021 (M.W.); the National Key Research and Development Program of China grants 2016YFC0105502 (L.L.), 2017YFA0205904 (B.H.), and 2017YFE0103600 (M.W.); Shenzhen International Cooperative Research Project grant GJHZ20180930110402104 (L.L.); Zhongyuan Thousand Talents Plan Project grant ZYQR201810117 (M.W.); NIH grants 1R01NS091604 (H.L.), R01DC017991 (H.L.), R21MH121831 (H.L.), and P50MH106435 (H.L.). L.D. is supported by a Canadian Institutes of Health Research postdoctoral fellowship, FRN: MFE-171291. We thank the National Center for Protein Sciences at Peking University for assistance with fMRI data processing. Data were provided in part by the Human Connectome Project, WU-Minn Consortium (Principal Investigators: David Van Essen and Kamil Ugurbil; 1U54MH091657) funded by the 16 NIH Institutes and Centers that support the NIH Blueprint for Neuroscience Research; and by the McDonnell Center for Systems Neuroscience at Washington University.

## Author contributions

H.L., Y.Y., B.H., L.L., and M.W. designed the research; Y.Y., L.D., J.R., L.S., X.P., R.W., C.H., C.J., C.G., Y.T., J.Z., Y.G., Y.L., and S.L. performed the research, Y.Y., L.D., J.R., and H.L. analyzed data, L.D., Y.Y., H.L., M.W., L.L., and B.H. wrote and improved the paper.

## Competing interests

Luming Li serves on the chief scientific advisory board for Beijing Pins Medical Co., Ltd, and is listed as an inventor on issued patents and patent applications pertaining to the deep-brain stimulator used in this work. The remaining authors declare no competing interests.
