## [Peer Review File · Nature Communications]

Reviewers' comments:

Reviewer #1 (Remarks to the Author):

Authors employed a newly-proposed deep machine learning model, called deep convolutional generative adversarial networks (DCGAN), to reconstructing lost BOLD signal in individual participants. They showed that the DCGAN model can be harnessed to learn individual patterns of brain activity and generate BOLD signals in artificially and non-artificially (i.e. DBS electrodes) compromised cortical regions.

1. Although interesting, I'm concerned about the real usefulness of this approach as - for instance - the artefact sizes are usually small in DBS patients and we can still have quite good results even with the artefact.

2. Additionally, it seems very difficult to create an algorithm replacing missing data. Authors gave the example of faces, which is fine since human faces have patterns, but do we know that BOLD signal has such patterns? Reconstructing in a healthy subject the BOLD signal with missing data is great, but I think that reconstructing a BOLD signal post DBS the same as pre DBS is flawed.... do we know that the signal has to be the same? wouldn't the lesioning effect change the BOLD signal? I think after the intervention it becomes hard to have a gold standard to compare to. Likewise, what if the DBS settings are changed (or DBS is simply turned on)? That would certainly change brain activity. In these cases the assumption that "functional brain organization, as assessed with functional connectivity, is assumed to be relatively stable over time" is simply not true.

Minor

3. please fix the reference "(Wang et al., 2015)", not quoted as a number.

Reviewer #2 (Remarks to the Author):

Reconstructing lost BOLD signal in individual participants using deep machine learning

The authors used GANs to reconstruct artificially lost signal in BOLD fMRI data. Time series correlated between original and reconstructed datasets at low- to moderate correlation coefficient strengths, and so did FC maps. The reconstruction technique was also applied to surgical patients with implanted depth electrodes.

The work has an interesting scope and the paper was overall well written. However, I was not so sure whether the authors made a strong case for the necessity and utility of their method, as i felt that their evaluations lacked the necessary rigor and depth. Their analytical pipeline introduces/enhances spatial autocorrelation at multiple levels, which may enhance performance artificially. Indeed, a simple learner that fills in signal from neighbouring non-missing vertices might perhaps perform similarly or even better than the proposed DNN model so one would like to see some more evaluations on that end. The evaluations were based on rather conventional and legacy 3T data (GSP); there was unfortunately no replication in a different, potentially more state of the art imaging dataset such as HCP, which may have been a worthwhile test of generalizability of the proposed technique. The clinical test case application was based on 2 patients only and thus relatively preliminary.

My main specific comments on the work are listed below:

1. The topic of correcting corrupted BOLD fMRI datasets is certainly a useful one. I was wondering, however, whether these complex models actually outperform simpler approaches and how good the performance really is given the somewhat low-to-medium correlation coefficients reported. Specific

comments relative to this issue can be seen below

a) The reconstruction technique fills in BOLD time series in cortical patches, and shows that simulated time series were "quite similar" i.e. correlate on average $\sim 0.23-0.35$ with actual time series. I was somewhat underwhelmed by the low strength of these correlations, and would think that these limit the utility of the technique in providing useable surrogates of lost signal. Please comment on this issue, and what a reasonable expectation is in this regard (also based on the suggested additional evaluations outlined below).

b) The GSP data that was used on the simulation experiment is somewhat conventional i.e. 3mm voxels and 3s TR. For a more technically oriented report like the current one, a more thorough evaluation across other datasets is strongly recommended. The HCP dataset, for example, offers approximately 2mm voxel size and sub-second temporal resolution; given the increasing shift towards multi-band acquisitions, HCP-grade scans becoming increasingly the state of the art at 3T and beyond. An additional evaluation of more "contemporary BOLD fMRI" data would strengthen the generalizability of the current work (and HCP data is already being provided in preprocessed form through the web).

c) Preprocessed data was smoothed on the cortical surface and then downsampled, which increases spatial autocorrelation between features and may thus favor prediction based on global signals. I was surprised that data were smoothed prior to parcellating them via the DK atlas, as this introduced additional data blurring.

d) Also with respect to the above comment: How well does a simple spatial diffusion of the boundary signals towards the missing vertices do (e.g. each missing vertex close to a boundary is assigned a weighted average of non-missing neighbouring vertices, and the process is iteratively performed until the missing patch is filled)? The authors are recommended to report mean correlations of such baseline models as well, to better evaluate the gain that can be observed from using more complex DNN approaches compared to simple in-painting approaches.

2. DK parcels are anatomically defined and do not necessarily respect functional boundaries, potentially blurring BOLD fMRI signals from different regions/networks. Other atlases, such as the Schaefer or Gordon parcellations respect functional boundaries more explicitly, and may thus provide more distinctive functional signals in specific ROIs. Evaluating the current approach based on these alternative atlases that may have more specific functional signatures would be important to provide additional information on the ability of the approach to fill in lost signal from specific functional clusters.

3. A further application would have been to also verify the ability of the approach in filling in missing TRs / time points. Some processing procedures such as scrubbing remove for example motion contaminated time points, so the scope of the current work could be broadened by evaluating whether the technique can adequately fill in missing TRs as well (at a whole-brain level).

4. I am not sure whether the comparison of preoperative to postoperative scans is a good evaluation scheme for the "in-painting" performance of the approach, given that there are inherent test-retest and state changes between both time points. Also, an evaluation based on only 2 patients seems somewhat preliminary to fully evaluate the model performance in a 'real world' case.

5. The authors carried out GSR during their preprocessing, which is a somewhat controversial technique in the overall rs-fMRI literature. How well does the approach perform when no GSR is

applied to the input data? Are findings overall robust and correlations similar? In the case of omission of GSR in the processing, how well does a simple in-painting technique perform that paints missing vertices with global signal, and how much better is the DNN compared to this (based on non-GSR corrected data) in terms of overall correlations?

Minor comment:

-Please specify how surface registration was done in the current work. Was this based on folding, or MSM-all? The latter may improve functional alignment, which may augment model performance.

Dear Editor,

Thank you for the opportunity to receive feedback on our manuscript, “*Reconstructing lost BOLD signal in individual participants using deep machine learning.*” We appreciate the thorough reviewer feedback – it has influenced our revised manuscript significantly. We have thought carefully about every point made, and thoroughly attempted to incorporate their implications into our work. Specifically, we have performed additional analyses suggested by reviewers and substantially increased our DBS patient sample size from 2 to 12 subjects and replicated the results. We believe that the result is a significantly stronger manuscript.

Although we believe that all concerns raised by reviewers are sufficiently addressed in the revised manuscript, we would be happy to make further revisions should further opportunities for improvement be apparent because our goal is to produce the strongest manuscript possible.

Below we list each reviewer’s comments and our response to each comment.

Reviewer #1 (Remarks to the Author):

Authors employed a newly-proposed deep machine learning model, called deep convolutional generative adversarial networks (DCGAN), to reconstructing lost BOLD signal in individual participants. They showed that the DCGAN model can be harnessed to learn individual patterns of brain activity and generate BOLD signals in artificially and non-artificially (i.e. DBS electrodes) compromised cortical regions.

1. Although interesting, I’m concerned about the real usefulness of this approach as - for instance - the artefact sizes are usually small in DBS patients and we can still have quite good results even with the artefact.

The reviewer is correct that *anatomical MRI* images may not be particularly compromised near the electrodes, however, we want to point out that *functional MRI signals* can be significantly compromised by wires outside the skull that connect the implanted electrodes to the stimulator. Please see raw fMRI data shown in Figure 4 (also shown below, compromised regions are indicated with arrows). The fMRI signal in some parts of the temporal lobe is almost completely lost. Moreover, we have observed that significant artifacts arise when the stimulator is turned on in the MRI scanner.

In the revised manuscript, we have supplemented the clinical portion of our study and added 10 patients with DBS electrodes to our analyses, totalling 12 patients. In these patients, we measured the surface area of the compromised region and, on average, the compromised area covered 8.36% of the cortical surface, which represents a substantial portion of the cortex (see Figures 4 and S3A for two examples). Among the 12 patients, the BOLD signal within the compromised area was nearly 40% lower on average compared to the presurgical BOLD signal (Figure S3B), with some regions showing a total loss of BOLD signal. Therefore, any study investigating functional connectivity or task-related BOLD activity in these patients will miss a large part of the picture if these artifacts are left unaddressed.

More importantly, the usefulness of our technique goes beyond the study on DBS patients. We want to emphasize that we simply used patients with DBS as an example to demonstrate the performance of our technique, because of the availability of pre-surgical and post-surgical data for comparisons. Once validated, the technique can be used in many other scenarios where fMRI signals are affected by different sources of noise. For example, MRI signals near the temporal pole and basal frontal lobe are known to be compromised by susceptibility artifacts. A technique that is capable of reconstructing lost signal at the individual subject level has the potential to address these long-time challenges in the field of MRI.

2. Additionally, it seems very difficult to create an algorithm replacing missing data. Authors gave the example of faces, which is fine since human faces have patterns, but do we know that BOLD signal has such patterns? Reconstructing in a healthy subject the BOLD signal with missing data is great, but I think that reconstructing a BOLD signal post DBS the same as pre DBS is flawed.... do we know that the signal has to be the same? wouldn't the lesioning effect change the BOLD signal? I think after the intervention it becomes hard to have a gold standard to compare to. Likewise, what if the DBS settings are changed (or DBS is simply turned on)?

That would certainly change brain activity. In these cases the assumption that “functional brain organization, as assessed with functional connectivity, is assumed to be relatively stable over time” is simply not true.

BOLD signals in different brain regions have very stable patterns when considered in relationship to one another. Based on these patterns, research over the last decade has identified multiple large-scale networks in the brain, whereby the activity of given sets of regions rise and descend together, signaling a cooperative functional connection between them (Biswal et al., 1995; Fox and Raichle, 2007; Greicius et al., 2003; Tononi et al., 1998; Van Dijk et al., 2010; Yeo et al., 2011). These networks are stable and are at play both during rest and during task performance (Bressler and Menon, 2010; Seeley et al., 2007; Toro et al., 2008) and give us insights into the functional organization of the human brain.

Regarding the justification of comparing post-DBS data to pre-DBS data, the reviewer is absolutely right that there will be differences between pre-surgical and post-surgical data, and there will be differences once the stimulator is turned on (although in this study we only used data when the stimulator is turned off). We would like to emphasize two important points. The first one is that the compromised region is not the same as the electrode implantation region. The electromagnetic interference occurs on the surface of the cortex that is adjacent to connection site on the skull. As such, while the electrode implantation site is in the subthalamic nucleus, the regions experiencing interference are located in the lateral parietal, lateral temporal, and lateral occipital cortices (see Figure 4, 5, S3, and S4). However, while the BOLD signal is disrupted in these patients due to the connectors, the underlying brain activity in these regions is intact. In other words, what is disrupted is the detection of the activity, not the activity itself. This leads us to our second point, which is that the functional connectivity of an intact region exhibits high stability over time, which renders possible the comparison of its activity at two timepoints. As a proof of concept, we investigated functional connectivity of the region that is contralateral to the compromised region, i.e. the right temporoparietal region, which suffered no interference. We found that the post-operative functional connectivity maps of the vertices within this right hemispheric region are highly and significantly correlated with their pre-operative functional connectivity maps ($r=0.75$, standard error=0.02; $t(10.98)=42.96$, $p<0.001$). As a visual reference, Figure S4 shows the functional connectivity maps of a given vertex in two patients, before and after electrode implantation (also shown below). The maps for the uncompromised region (right-hand column) exhibit very high similarity: In Patient 1, the correlation between the functional connectivity maps at the two different timepoints is $r = 0.81$, while in Patient 2 the correlation is $r = 0.85$. This proof of concept directly demonstrates the viability of comparing pre- and post-implantation fMRI data to assess the accuracy of our reconstruction method.

Fig. S4. FC maps remain stable after electrode implantation.

Here we show proof of concept that FC maps remain stable following electrode implantation surgery. We generated pre- and post-operative FC maps from one seed in the compromised region (left hemisphere), and the same seed in the uncompromised region in the right hemisphere. FC maps are shown here for two representative patients. The left hemisphere post-operative FC map presents substantial differences from the pre-operative FC map, as indicated by a low correlation of $r = 0.28$ in Patient 1 and $r = 0.47$ in Patient 2. When looking at the uncompromised seed in the right hemisphere, the post-operative FC map is highly similar to the intact pre-operative FC map, as shown by a correlation of $r = 0.81$ in Patient 1 and $r = 0.85$ in Patient 2. Thus, functional connectivity does not seem substantially affected by electrode implantation.

These findings were added to the manuscript:

“However, functional brain organization, as assessed with functional connectivity, is assumed to be relatively stable over time¹⁵. Although the implantation surgery may cause microlesion effects and lead to changes in brain circuits involving the stimulation target (i.e., the subthalamic nucleus), the surgery is less likely to change functional connectivity in the area of signal loss, which is relatively far from the location of the stimulator and the motor network being modulated. As a proof of concept, we investigated functional connectivity of the right temporoparietal region of the two patients (note that signal loss was observed only in the left hemisphere). We found that the post-operative FC maps in the right hemisphere are highly and significantly correlated with the pre-operative FC maps ($r=0.75$, standard error=0.02; $t(10.98)=42.96$, $p<0.001$). However, for the compromised region in the left hemisphere, we found that the post-operative FC maps are only weakly positively correlated with the pre-operative FC maps ($r=0.37$, standard error=0.03; $t(11.02)=11.24$, $p<0.001$). As an example, we show cortical FC maps using a seed placed in two representative patients’ compromised regions (Fig. S4). Unlike the weak and disorganized FC maps obtained from patients’ compromised left temporoparietal region, the FC maps generated from seeds in the right temporoparietal region show high similarity to their pre-operative FC maps (right hemisphere seeds across the two

patients in Figure S4: $r=0.83\pm0.03$, $p<0.001$; left hemisphere seeds across both patients: $r=0.38\pm0.13$, $p<0.001$) (Fig. S4).

Having shown that FC maps are relatively stable following electrode implantation, we next assessed the reconstructive accuracy of our DCGAN model for the patients' compromised region in the left hemisphere."

Minor

3. please fix the reference "(Wang et al., 2015)", not quoted as a number.

Thank you for pointing this out. We have fixed the reference.

Reviewer #2 (Remarks to the Author):

The authors used GANs to reconstruct artificially lost signal in BOLD fMRI data. Time series correlated between original and reconstructed datasets at low- to moderate correlation coefficient strengths, and so did FC maps. The reconstruction technique was also applied to surgical patients with implanted depth electrodes.

The work has an interesting scope and the paper was overall well written. However, I was not so sure whether the authors made a strong case for the necessity and utility of their method, as I felt that their evaluations lacked the necessary rigor and depth. Their analytical pipeline introduces/enhances spatial autocorrelation at multiple levels, which may enhance performance artificially. Indeed, a simple learner that fills in signal from neighbouring non-missing vertices might perhaps perform similarly or even better than the proposed DNN model so one would like to see some more evaluations on that end. The evaluations were based on rather conventional and legacy 3T data (GSP); there was unfortunately no replication in a different, potentially more state of the art imaging dataset such as HCP, which may have been a worthwhile test of generalizability of the proposed technique. The clinical test case application was based on 2 patients only and thus relatively preliminary.

My main specific comments on the work are listed below:

1. The topic of correcting corrupted BOLD fMRI datasets is certainly a useful one. I was wondering, however, whether these complex models actually outperform simpler approaches and how good the performance really is given the somewhat low-to-medium correlation coefficients reported. Specific comments relative to this issue can be seen below

- a) The reconstruction technique fills in BOLD time series in cortical patches, and shows that simulated time series were "quite similar" i.e. correlate on average $\sim 0.23-0.35$ with actual time series. I was somewhat underwhelmed by the low strength of these correlations, and would think that these limit the utility of the technique in providing useable surrogates of lost signal. Please comment on this issue, and what a reasonable expectation is in this regard (also based on the suggested additional evaluations outlined below).

We would like to thank the reviewer for raising this important point. We feel it necessary here to make a distinction between BOLD time series and functional connectivity derived from them. We reconstructed the BOLD images frame by frame without using any temporal information and then checked the BOLD time series *post hoc*. While the reconstructed BOLD signal showed moderate correlation with original intact signal, functional connectivity patterns derived from these signals are more amenable to interpretation, and carry more useful information for basic and clinical applications. In our study we found that functional connectivity maps derived from reconstructed vertices exhibited average correlations between $r = 0.61$ and $r = 0.79$ with the original intact connectivity maps, which are considered very high correlations. In a previous publication that systematically evaluated the test-retest reliability of resting state functional

connectivity, it has been shown that the overall test-retest reliability is between 0.5 and 0.6 with 12-min fMRI data (see Figure 5 in Mueller et al., 2015). *This suggests that the reliability of functional connectivity maps derived from our reconstructed BOLD signal is considerably higher than test-retest reliability; in other words, even if one can fix the noise source that causes the BOLD signal loss and rescan the subject, the result is no better than using our technique to reconstruct the compromised signal.*

Another important point that we would like to emphasize is the individual-specific nature of the reconstructions. The reconstructed functional connectivity maps are most similar to the subject's own intact maps rather than to the group map or the most similar individual's map. Therefore, the reconstructed signals carry individual-specific information, which is crucial for clinical applications where patient heterogeneity requires researchers to move away from group-level analyses.

b) The GSP data that was used on the simulation experiment is somewhat conventional i.e. 3mm voxels and 3s TR. For a more technically oriented report like the current one, a more thorough evaluation across other datasets is strongly recommended. The HCP dataset, for example, offers approximately 2mm voxel size and sub-second temporal resolution; given the increasing shift towards multi-band acquisitions, HCP-grade scans becoming increasingly the state of the art at 3T and beyond. An additional evaluation of more "contemporary BOLD fMRI" data would strengthen the generalizability of the current work (and HCP data is already being provided in preprocessed form through the web).

We agree that it is necessary to test the success of the model in different datasets to demonstrate generalizability. To this end, we analyzed healthy young adult data acquired through a Siemens MRI scanner (the GSP data), and older adult patient data acquired through a Philips scanner (the DBS data), with different MRI sequences. We observed good performance of our technique in both datasets. Importantly, the patient data was of lower quality, mostly due to increased head motion in the patient sample. *We chose to test our method in these lower-quality datasets as they better represent real-world clinical applications than the high-quality HCP data acquired using the state-of-the-art PRISMA scanner.* It goes without saying that it is a harder task to generate accurate reconstructions with data that is of lower quality. Therefore, we do not believe that using state-of-the-art fMRI data would add any valuable information above and beyond what we have achieved in these two datasets. If anything, we hypothesize that the reconstructions might be even more accurate, as state-of-the-art data would have, as the reviewer mentioned, shorter TRs, and therefore more volumes and more data for a given scan duration. Feeding more data to a machine learning algorithm will always output a better model, and therefore more accurate reconstructions.

In summary, the success of our model in two datasets that differ in MRI scanners, MRI sequences, head motion, age, and health status support the generalizability of the machine learning method presented in the paper.

c) Preprocessed data was smoothed on the cortical surface and then downsampled, which

increases spatial autocorrelation between features and may thus favor prediction based on global signals. I was surprised that data were smoothed prior to parcellating them via the DK atlas, as this introduced additional data blurring.

While spatial autocorrelation is a concern in fMRI research, most groups still choose to apply smoothing to resting-state fMRI data as the benefit of boosting SNR is greater than the cost of maintaining autocorrelations. To our knowledge, almost all parcellation methods use smoothed data, including cortical parcellations such as Glasser et al. (2016); Gordon et al. (2016); Schaefer et al. (2018), as well as parcellations of smaller regions such as the hippocampus, like Plachti et al. (2019) and Robinson et al. (2015). Smoothing serves to lessen partial volume effects and boost signal-to-noise ratio, which is useful for parcellation. Even state-of-the-art methods, such as Glasser's multi-modal parcellation (Glasser et al., 2016), uses data that is at least minimally smoothed before parcellation. Glasser et al. were able to minimize smoothing as they used an areal parcellation.

Importantly, in our study the DKT atlas was only used to define separate regions to be artificially compromised. For example, we used DKT labels 4, 13, 19, 20, 21, 28 to define the lateral frontal cortex, in which we subsequently removed the BOLD signal. However, the data fed to the machine learning algorithm for training, as well as the data it outputted during testing, was in vertex space. The reconstruction of the BOLD signal was therefore done at the level of individual vertices within the regions, rather than at the level of the DKT-based region as a whole. As such, there was no additional blurring of the data through DKT segmentation.

We have added clarifications regarding the use of vertex space by the machine learning algorithm in the Results section:

*“The DCGAN model was trained on a resting-state fMRI dataset of 80 randomly-chosen participants (240 frames for each participant, **in vertex space**) from the publicly-available Brain Genomics Superstruct Project (GSP) database¹². In an independent test dataset comprised of 20 participants, we artificially removed BOLD signal in various cortical surface regions and used the trained generator to reconstruct compromised BOLD frames **in vertex space**.”*

Methods section:

*“Eighty participants were randomly selected from the Brain GSP dataset¹² to build the training dataset, which consisted of 19,200 intact flattened BOLD frames **in vertex space**. The data from 20 other participants, again selected randomly from the GSP, constituted the test dataset, which was independent from the training dataset. We used a DCGAN model² to create BOLD frames **in vertex space** based on each individual participant's data.”*

d) Also with respect to the above comment: How well does a simple spatial diffusion of the boundary signals towards the missing vertices do (e.g. each missing vertex close to a boundary is assigned a weighted average of non-missing neighbouring vertices, and the process is iteratively performed until the missing patch is filled)? The authors are recommended to report mean correlations of such baseline models as well, to better evaluate the gain that can be observed from using more complex DNN approaches compared to simple in-painting approaches.

The reviewer suggested an interesting spatial diffusion method. However, a simple linear interpolation method implies an inherent homogeneity within the compromised region, and will inevitably result in a smoothed patch (because it is essentially a weighted smoothing approach). This will lead to high functional correlations within the patch, and high correlations between the filled area and neighboring regions. Secondly, as can be seen in the real-world example of patients with intracortical electrodes (see Figure 5), the compromised region is often quite large. If we use the spatial diffusion method, signal reconstruction of the vertices on the border surrounding the compromised region will be more reasonable but the vertices inside the perimeter would get progressively worse as we move further away from the perimeter and towards the center. The region of signal loss may comprise vertices that have very different functional connectivity profiles. This is evidenced by parcellation methods such as Glasser’s, Schaefer’s, our own (Wang et al., 2015) and others’, which all show that the area encompassing the lateral temporal, parietal, and occipital cortices is made up of many different functional networks. A spatial diffusion method would not allow for sharp differentiations in functional connectivity or BOLD activity between neighboring voxels or vertices and would instead mandate gradients.

However, as can be seen in Figure 1 (also shown below), our machine learning method is able to reconstruct largely uncorrelated signals in two proximal vertices, that also have very different functional connectivity profiles. Critically, the reconstructed signal can even recover anti-correlations between the compromised region and its neighboring regions (see Vertex 2 below and its anti-correlation with the adjacent temporoparietal junction). Complex, fine-grained functional connectivity patterns within the compromised regions were also reconstructed in the DBS patients, which may not have been recovered if a spatial smoothing approach had been used.

2. DK parcels are anatomically defined and do not necessarily respect functional boundaries,

potentially blurring BOLD fMRI signals from different regions/networks. Other atlases, such as the Schaefer or Gordon parcellations respect functional boundaries more explicitly, and may thus provide more distinctive functional signals in specific ROIs. Evaluating the current approach based on these alternative atlases that may have more specific functional signatures would be important to provide additional information on the ability of the approach to fill in lost signal from specific functional clusters.

We apologize for the unclear description of the method that might have caused this confusion. As mentioned above, the DKT parcels were only used to define regions within which we subsequently removed the signal in order to artificially compromise different regions throughout the cortical surface. Technically, we can remove signal from any arbitrary patch or use any anatomical or functional atlas. The reconstruction of the signal was done at the level of individual vertices, and functional connectivity was measured across vertices and not across DKT parcels. Therefore, our machine learning method was not subject to any bias introduced by the atlas. This has been clarified in the revised manuscript (see added text in response to comment 1c).

3. A further application would have been to also verify the ability of the approach in filling in missing TRs / time points. Some processing procedures such as scrubbing remove for example motion contaminated time points, so the scope of the current work could be broadened by evaluating whether the technique can adequately fill in missing TRs as well (at a whole-brain level).

We would like to thank the reviewer for proposing this idea and we completely agree that it would be extremely valuable if the method can in the future be extended to filling in missing TRs. If successful, this would address the whole-brain signal contamination problem caused by head motion and make a major contribution to the field of fMRI. However, interpolating TRs is a much harder problem that goes beyond the scope of current study, in that one must consider not only the spatial information within the frame but also the complex temporal patterns between frames. In other words, the model must be able to learn not only the 2-D spatial patterns but also the 3-D temporal dynamics. Thus, the complexity of learning will be exponentially increased.

Nevertheless, given the preliminary success in recovering signal within the frame, we are enthusiastic about the prospect of using machine learning for the important applications proposed by the reviewer. We have included this future direction in the Discussion.

4. I am not sure whether the comparison of preoperative to postoperative scans is a good evaluation scheme for the "in-painting" performance of the approach, given that there are inherent test-retest and state changes between both time points. Also, an evaluation based on only 2 patients seems somewhat preliminary to fully evaluate the model performance in a 'real world' case.

We completely agree with the reviewer that our patient sample was too preliminary. In the revised manuscript we have substantially enlarged our sample size and included 12 surgical patients who were implanted with deep brain stimulators (DBS). The results are largely the same

as those reported previously with two patients. These findings demonstrate that our method can successfully be used in a real-world clinical study.

Regarding the justification of comparing post-DBS data to pre-DBS data, the reviewer is absolutely right that there will be differences between pre-surgical and post-surgical data, and there will be differences once the stimulator is turned on. We would like to emphasize two important points. The first one is that the compromised region is not the same as the electrode implantation region. The electromagnetic interference occurs on the surface of the cortex that is adjacent to connection site on the skull. As such, while the electrode implantation site is in the subthalamic nucleus, the regions experiencing interference are located in the lateral parietal, lateral temporal, and lateral occipital cortices (see Figure 4, 5, S3, and S4). However, while the BOLD signal is disrupted in these patients due to the connectors, the underlying brain activity in these regions is intact. In other words, what is disrupted is the detection of the activity, not the activity itself.

This leads us to our second point, which is that the functional connectivity of an intact region exhibits high stability over time, which renders possible the comparison of its activity at two timepoints. As a proof of concept, we investigated the functional connectivity of the region that is contralateral to the compromised region in two patients, i.e. the right temporoparietal region, which suffered no interference. We found that the post-operative functional connectivity maps of the vertices within this right hemispheric region are highly and significantly correlated with their pre-operative functional connectivity maps ($r=0.75$, standard error=0.02; $t(10.98)=42.96$, $p<0.001$). As a visual reference, Figure S4 shows the functional connectivity maps of a given vertex in two patients, before and after electrode implantation (also shown below). The maps for the uncompromised region (right-hand column) exhibit very high similarity: In Patient 1, the correlation between the functional connectivity maps at the two different timepoints is $r = 0.81$, while in Patient 2 the correlation is $r = 0.85$. This proof of concept directly demonstrates the viability of comparing pre- and post-implantation fMRI data to assess the accuracy of our reconstruction method.

Fig. S4. FC maps remain stable after electrode implantation.

Here we show proof of concept that FC maps remain stable following electrode implantation surgery. We generated pre- and post-operative FC maps from one seed in the compromised region (left hemisphere), and the same seed in the uncompromised region in the right hemisphere. FC maps are shown here for two representative patients. The left hemisphere post-operative FC map presents substantial differences from the pre-operative FC map, as indicated by a low correlation of $r = 0.28$ in Patient 1 and $r = 0.47$ in Patient 2. When looking at the uncompromised seed in the right hemisphere, the post-operative FC map is highly similar to the intact pre-operative FC map, as shown by a correlation of $r = 0.81$ in Patient 1 and $r = 0.85$ in Patient 2. Thus, functional connectivity does not seem substantially affected by electrode implantation.

These findings were added to the manuscript:

“However, functional brain organization, as assessed with functional connectivity, is assumed to be relatively stable over time¹⁵. Although the implantation surgery may cause microlesion effects and lead to changes in brain circuits involving the stimulation target (i.e., the subthalamic nucleus), the surgery is unlikely to change functional connectivity in the area of signal loss, which is relatively far from the location of the stimulator and the motor network being modulated. As a proof of concept, we investigated functional connectivity of the right temporoparietal region of the two patients (note that signal loss was observed only in the left hemisphere). We found that the post-operative FC maps in the right hemisphere are highly and significantly correlated with the pre-operative FC maps ($r=0.75$, standard error=0.02; $t(10.98)=42.96$, $p<0.001$). However, for the compromised region in the left hemisphere, we found that the post-operative FC maps are only weakly positively correlated with the pre-operative FC maps ($r=0.37$, standard error=0.03; $t(11.02)=11.24$, $p<0.001$). As an example, we show cortical FC maps using a seed placed in two representative patients’ compromised regions (Fig. S4). Unlike the weak and disorganized FC maps obtained from patients’ compromised left temporoparietal region, the FC maps generated from seeds in the right temporoparietal region show high similarity to their pre-operative FC maps (right hemisphere seeds across the two patients in Figure S4: $r=0.83\pm0.03$, $p<0.001$; left hemisphere seeds across both patients: $r=0.38\pm0.13$, $p<0.001$) (Fig. S4).

Having shown that FC maps are relatively stable following electrode implantation, we next assessed the reconstructive accuracy of our DCGAN model for the patients’ compromised region in the left hemisphere.”

5. The authors carried out GSR during their preprocessing, which is a somewhat controversial technique in the overall rs-fMRI literature. How well does the approach perform when no GSR is applied to the input data? Are findings overall robust and correlations similar? In the case of omission of GSR in the processing, how well does a simple in-painting technique perform that paints missing vertices with global signal, and how much better is the DNN compared to this (based on non-GSR corrected data) in terms of overall correlations?

As the reviewer duly noted, GSR has been a topic of debate for over a decade. The major problem associated with GSR is that it introduces spurious temporal anticorrelations in functional connectivity analyses. To date, there is still no consensus about whether GSR should or should not be included in resting state data processing. The benefits and disadvantages of including GSR may also vary depending on the purpose of the studies. For example, Power et al. (2017b) compared various combinations of denoising strategies. They showed that GSR is the only denoising method that effectively removes global signals, including artefactual signals and, crucially, global neural activity (Power et al., 2017b), which is proposed to be modulated mainly by respiration and/or arousal/vigilance (falling asleep is linked to slower and deeper breathing (Power et al., 2017a; Power et al., 2017b)). GSR is thus useful in studies that wish to control for global neural activity (e.g., controlling for levels of alertness), but should be avoided in studies in which the global neural signal is of interest, as for example in arousal or sleep studies (Power et al., 2017a). In a recent study, (Li et al., 2019) highlighted the usefulness of GSR in detecting individual differences in behavior; resting-state data preprocessed with GSR led to greater associations between whole-brain functional connectivity and measures of cognition, personality, and emotion. Whether GSR should be used for resting state fMRI preprocessing is beyond the scope of our study. However, given that GSR is useful in controlling for global noise, we chose to include this step in this study. It is worth noting that our image reconstruction was performed in each frame independently of the temporal correlations. Thus, the inclusion or exclusion of GSR would have little impact on the reconstruction technique.

Minor comment:

-Please specify how surface registration was done in the current work. Was this based on folding, or MSM-all? The latter may improve functional alignment, which may augment model performance.

The registration was based on folding. We completely agree with the reviewer that MSMAll improves functional alignment. MSMAll is an inter-subject registration based on a multimodal surface matching (MSM) algorithm and a variety of modalities released by the HCP (Glasser et al., 2016; Robinson et al., 2014). In our previous work, we have evaluated functional connectivity measures based on MSM-all registration and confirmed that it outperforms folding-based registration (Li et al., 2019). However, MSMAll registration requires multi-modal features such as myelin maps, resting state network maps, and resting-state visuotopic maps (Glasser et al., 2016). In this study, our data are BOLD images with compromised signals, thus it is probably invalid to use the compromised information for registration.

References

- Biswal, B, Zerrin Yetkin, F, Haughton, VM, and Hyde, JS (1995). Functional connectivity in the motor cortex of resting human brain using echo - planar MRI. *Magnetic resonance in medicine*, 34, 537-541.
- Bressler, SL, and Menon, V (2010). Large-scale brain networks in cognition: emerging methods and principles. *Trends in cognitive sciences*, 14, 277-290.
- Fox, MD, and Raichle, ME (2007). Spontaneous fluctuations in brain activity observed with functional magnetic resonance imaging. *Nature reviews neuroscience*, 8, 700-711.
- Glasser, MF, Coalson, TS, Robinson, EC, Hacker, CD, Harwell, J, Yacoub, E, Ugurbil, K, Andersson, J, Beckmann, CF, and Jenkinson, M (2016). A multi-modal parcellation of human cerebral cortex. *Nature*, 536, 171-178.
- Gordon, EM, Laumann, TO, Adeyemo, B, Huckins, JF, Kelley, WM, and Petersen, SE (2016). Generation and evaluation of a cortical area parcellation from resting-state correlations. *Cerebral cortex*, 26, 288-303.
- Greicius, MD, Krasnow, B, Reiss, AL, and Menon, V (2003). Functional connectivity in the resting brain: a network analysis of the default mode hypothesis. *Proceedings of the National Academy of Sciences*, 100, 253-258.
- Li, M, Wang, D, Ren, J, Langs, G, Stoecklein, S, Brennan, BP, Lu, J, Chen, H, and Liu, H (2019). Performing group-level functional image analyses based on homologous functional regions mapped in individuals. *PLoS biology*, 17, e2007032.
- Mueller, S, Wang, D, Fox, MD, Pan, R, Lu, J, Li, K, Sun, W, Buckner, RL, and Liu, H (2015). Reliability correction for functional connectivity: Theory and implementation. *Human brain mapping*, 36, 4664-4680.
- Plachti, A, Eickhoff, SB, Hoffstaedter, F, Patil, KR, Laird, AR, Fox, PT, Amunts, K, and Genon, S (2019). Multimodal parcellations and extensive behavioral profiling tackling the hippocampus gradient. *Cerebral cortex*, 29, 4595-4612.
- Power, JD, Laumann, TO, Plitt, M, Martin, A, and Petersen, SE (2017a). On global fMRI signals and simulations. *Trends in cognitive sciences*, 21, 911-913.
- Power, JD, Plitt, M, Laumann, TO, and Martin, A (2017b). Sources and implications of whole-brain fMRI signals in humans. *Neuroimage*, 146, 609-625.
- Robinson, EC, Jbabdi, S, Glasser, MF, Andersson, J, Burgess, GC, Harms, MP, Smith, SM, Van Essen, DC, and Jenkinson, M (2014). MSM: a new flexible framework for Multimodal Surface Matching. *Neuroimage*, 100, 414-426.
- Robinson, JL, Barron, DS, Kirby, LA, Bottenhorn, KL, Hill, AC, Murphy, JE, Katz, JS, Salibi, N, Eickhoff, SB, and Fox, PT (2015). Neurofunctional topography of the human hippocampus. *Human brain mapping*, 36, 5018-5037.
- Schaefer, A, Kong, R, Gordon, EM, Laumann, TO, Zuo, X-N, Holmes, AJ, Eickhoff, SB, and Yeo, BT (2018). Local-global parcellation of the human cerebral cortex from intrinsic functional connectivity MRI. *Cerebral Cortex*, 28, 3095-3114.
- Seeley, WW, Menon, V, Schatzberg, AF, Keller, J, Glover, GH, Kenna, H, Reiss, AL, and Greicius, MD (2007). Dissociable intrinsic connectivity networks for salience processing and executive control. *Journal of Neuroscience*, 27, 2349-2356.
- Tononi, G, McIntosh, AR, Russell, DP, and Edelman, GM (1998). Functional clustering: identifying strongly interactive brain regions in neuroimaging data. *Neuroimage*, 7, 133-149.
- Toro, R, Fox, PT, and Paus, T (2008). Functional coactivation map of the human brain. *Cerebral cortex*, 18, 2553-2559.

- Van Dijk, KR, Hedden, T, Venkataraman, A, Evans, KC, Lazar, SW, and Buckner, RL (2010). Intrinsic functional connectivity as a tool for human connectomics: theory, properties, and optimization. *Journal of neurophysiology*, 103, 297-321.
- Wang, D, Buckner, RL, Fox, MD, Holt, DJ, Holmes, AJ, Stoecklein, S, Langs, G, Pan, R, Qian, T, and Li, K (2015). Parcellating cortical functional networks in individuals. *Nature neuroscience*, 18, 1853-1860.
- Yeo, B, Krienen, FM, Sepulcre, J, Sabuncu, MR, Lashkari, D, Hollinshead, M, Roffman, JL, Smoller, JW, Zöllei, L, and Polimeni, JR (2011). The organization of the human cerebral cortex estimated by intrinsic functional connectivity. *Journal of neurophysiology*, 106, 1125-1165.

REVIEWER COMMENTS

Reviewer #1 (Remarks to the Author):

I'm satisfied with the new version of the manuscript.

Reviewer #2 (Remarks to the Author):

I would like to thank the authors for their response. However, I had the impression that several of my more substantive comments were not thoroughly addressed in this revision. Some analyses were suggested to strengthen the current study, but these were not carried out by the authors unfortunately.

1) With respect to my previous suggestion 1b, the authors reply that it is an easier task to reconstruct a high resolution signal compared to a low resolution one and thus do not carry out the evaluation based on the the HCP dataset (which the authors do not demonstrate however). I would have actually predicted the opposite, as a high resolution signal contains more high frequency content. Moreover, rather 'naive' reconstruction techniques based on blurring may not achieve a high performance for high resolution data while they still fare ok for low def data.

I would thus like to reiterate my suggestion that the authors should ideally evaluate their algorithm for HCP-style data as well. HCP is an open and easily accessible benchmark dataset that is widely used in the neuroimaging community. Furthermore, and unlike the authors' statement, HCP style acquisitions are increasingly becoming the state-of-the-art for clinical research as well (see eg. also the disease-oriented initiatives from HCP), so showing generalizability of their method to HCP type datasets would be worthwhile. As such, I think the works scope can be increased by evaluating how well the algorithm works for data that has higher spatial / temporal resolution.

2) In their response to my comment 1d), where I asked about the performance benefits relative to a simple diffusion methods, the authors give a conceptual response. While this is fair, my comment was trying to encourage the authors to run the diffusion method as a baseline model that they can compare the performance of their neural network to. This evaluation would then allow them to provide an answer that is supported by this assessment, and in particular whether their approach outperforms rather 'naive' inpainting methods that simply operate based on tissue blurring.

3) My prior question 5) asked about the stability of their algorithm relative to including/omitting GSR. This was not fully answered in the authors response.

Response to reviewers' comments

Reviewer #1 (Remarks to the Author):

I'm satisfied with the new version of the manuscript.

We would like to thank the reviewer for taking the time to review our manuscript.

Reviewer #2 (Remarks to the Author):

I would like to thank the authors for their response. However, I had the impression that several of my more substantive comments were not thoroughly addressed in this revision. Some analyses were suggested to strengthen the current study, but these were not carried out by the authors unfortunately.

We thank the reviewer for the careful and thoughtful review of our manuscript and the highly constructive comments. We apologize for the initial incomplete response, but are happy to let the reviewer know that we have now performed all the analyses requested by the reviewer, and we feel that these additional findings make the manuscript substantially stronger and more robust than the initial version. We hope that the reviewer will find these new analyses satisfactory.

1) With respect to my previous suggestion 1b, the authors reply that it is an easier task to reconstruct a high resolution signal compared to a low resolution one and thus do not carry out the evaluation based on the the HCP dataset (which the authors do not demonstrate however). I would have actually predicted the opposite, as a high resolution signal contains more high frequency content. Moreover, rather 'naive' reconstruction techniques based on blurring may not achieve a high performance for high resolution data while they still fare ok for low def data.

I would thus like to reiterate my suggestion that the authors should ideally evaluate their algorithm for HCP-style data as well. HCP is an open and easily accessible benchmark dataset that is widely used in the neuroimaging community. Furthermore, and unlike the authors' statement, HCP style acquisitions are increasingly becoming the state-of-the-art for clinical research as well (see eg. also the disease-oriented initiatives from HCP), so showing generalizability of their method to HCP type datasets would be worthwhile. As such, I think the works scope can be increased by evaluating how well the algorithm works for data that has higher spatial / temporal resolution.

We agree that the widespread use of the HCP dataset makes it especially relevant to test our model on these data, and have performed the analyses requested by the reviewer. For each HCP participant, we truncated the data to 8 min to match the GSP dataset. We had initially hypothesized that reconstructive accuracy would be higher for HCP data due to its higher spatial and temporal resolution. However, as the reviewer correctly predicted, we found the opposite. Nevertheless, our model still successfully reconstructed HCP time series and functional connectivity (FC) maps in all cortical regions (see the new Table 1 below for detailed results on each cortical region), the time series reconstructions were marginally less accurate than the GSP-based ones after Bonferroni correction (independent samples t-test on data collapsed across cortical regions, time series: $t(30.21)=2.41$, $p=0.022$; significant p threshold = 0.017) and the FC maps reconstructions were significantly less accurate ($t(38)=12.88$, $p<0.001$) (new Fig. 4A).

Because our model is learning the spatial patterns embedded in each individual frame of the fMRI data, we postulated that signal-to-noise ratio of the images would have a significant impact on the

model performance. We thus compared the temporal SNR (tSNR) of HCP data and GSP data. We calculated tSNR based on the standard deviation of the BOLD signal (Triantafyllou et al., 2005; Marcus et al., 2013), and found that the HCP dataset indeed has a significantly lower tSNR than the GSP dataset (independent samples t-test, $t(139.43)=49.31$, bootstrapped $p = 0.001$) (new Fig. 4B).

To demonstrate that a higher SNR of the fMRI data will lead to increased model performance, we temporally smoothed the BOLD time series of the HCP data by averaging every 4 images together. This temporal smoothing procedure is equivalent to increasing the TR of HCP data from 0.72s to 2.88 s ($0.72 \text{ s} \times 4$), which is comparable to the GSP TR (3.00 s). We found that the tSNR was significantly improved compared to the raw data (paired samples t-test, $t(99)=-40.68$, $p < 0.001$) (new Fig.4B). We then used the temporally smoothed HCP data for model training and testing, and found that the reconstructions were more accurate than those of the model trained on raw HCP data (paired samples t-test on data collapsed across all cortical regions, time series: $t(19)=-4.67$, $p<0.001$; FC maps: $t(19)=143.21$, $p<0.001$) (new Fig. 4A; see Table 1 for reconstructive accuracy of each model in the five cortical regions). Compared to the GSP dataset, the temporally smoothed HCP data still had lower tSNR (independent samples t-test, $t(198)=23.26$, bootstrapped $p = 0.001$) (new Fig. 4B), however its reconstructions were non-significantly different from the GSP's in terms of time series (independent samples t-test: $t(32.12)=1.16$, $p = 0.26$), and marginally less accurate in terms of FC maps after Bonferroni correction (independent samples t-test, $t(38)=2.43$, $p = 0.02$; significant p threshold=0.017). These results indicate that SNR of the training images is an important factor to consider for machine learning-based reconstruction.

We have added the following text and figures to the manuscript:

Results:

“We replicated our findings by performing the same time series and FC-based analyses in a dataset sporting higher spatial and temporal resolutions: the HCP dataset¹². Again, the data of 80 randomly chosen participants were used to train our DCGAN model, and the data of 20 independent participants were used to test reconstruction. Multilevel linear models revealed significant positive correlations between the original and reconstructed time series in all regions: temporal cortex: $r=0.31\pm0.05$ (CI [0.29, 0.33], $t(19)=27.60$, $p<0.001$), lateral frontal cortex: $r=0.12\pm0.04$ (CI [0.11, 0.14], $t(19)=14.33$, $p<0.001$), medial frontal cortex: $r=0.18\pm0.05$ (CI [0.16, 0.20], $t(19)=16.73$, $p<0.001$), parietal cortex: $r=0.24\pm0.05$ (CI [0.22, 0.27], $t(19)=19.71$, $p<0.001$), and occipital cortex: $r=0.35\pm0.07$ (CI [0.32, 0.39], $t(19)=23.39$, $p<0.001$) (Table 1). We reconstructed FC maps, and multilevel linear models again revealed significant positive correlations in all regions: temporal cortex: $r=0.60\pm0.07$ (CI [0.57, 0.63], $t(19)=38.04$, $p<0.001$), lateral frontal cortex: $r=0.44\pm0.05$ (CI [0.42, 0.47], $t(19)=39.47$, $p<0.001$), medial frontal cortex: $r=0.46\pm0.06$ (CI [0.43, 0.49], $t(19)=32.53$, $p<0.001$), parietal cortex: $r=0.62\pm0.03$ (CI [0.60, 0.63], $t(19)=81.92$, $p<0.001$), and occipital cortex: $r=0.65\pm0.06$ (CI [0.62, 0.68], $t(19)=46.90$, $p<0.001$) (Table 1). Comparing the HCP and GSP reconstructions, the GSP-trained model yielded marginally more accurate time series reconstructions when correcting for multiple comparisons (mean difference=0.02, CI [0.004, 0.04], $t(30.21)=2.41$, $p=0.022$, $\eta^2=0.16$; significant p threshold=0.017) and significantly more accurate FC map reconstructions (mean difference=0.11, CI [0.10, 0.13], $t(38)=12.88$, $p<0.001$, $\eta^2=0.81$) (Fig. 4A), despite the HCP dataset having higher spatial and temporal resolutions. We postulated that the HCP data may have a lower temporal signal-to-noise ratio (tSNR) than the GSP dataset, and indeed this was the case ($t(139.43)=49.31$, bootstrapped $p=0.001$, $\eta^2=0.95$) (Fig. 4B). To counteract this, we temporally smoothed the HCP data by averaging together every 4 frames before retraining and retesting the model. This significantly increased tSNR ($t(99)=-40.68$, $p<0.001$, $\eta^2=0.94$) (Fig. 4B) and yielded more accurate reconstructions than the raw HCP-trained model (time series: mean

difference=-0.01, CI [-0.02, -0.01], $t(19)=-4.67$, $p<0.001$, $\eta^2=0.53$; FC maps: mean difference=-0.09, CI [-0.10, -0.09], $t(19)=-143.21$, $p<0.001$; $\eta^2=1.00$) (Fig. 4A; see Table 1 for reconstruction details on each cortical region). The tSNR of the temporally smoothed HCP data remained lower than the GSP's ($t(198)=23.26$, bootstrapped $p=0.001$, $\eta^2=0.73$) (Fig. 4B), however the time series reconstructions are similar in accuracy to the GSP's (mean difference=0.01, CI [-0.008, 0.03], $t(32.12)=1.16$, $p=0.26$, $\eta^2=0.04$), while the reconstructed FC maps are marginally less accurate after Bonferroni correction (mean difference=0.02, CI [0.003, 0.04], $t(38)=2.43$, $p=0.020$, $\eta^2=0.13$; significant p threshold=0.017) (Fig. 4A). These findings indicate that tSNR has an important effect on machine learning reconstructive accuracy.”

Discussion:

“Of note, the temporal signal-to-noise ratio (SNR) of the training data is an important factor that modulates reconstructive accuracy.”

“We have replicated our findings in multiple datasets and have shown that it is possible to reconstruct lost BOLD signal in healthy individuals as well as in a clinical sample with compromised fMRI.”

Table 1. Time series and functional connectivity map reconstructive accuracy following various models.

Reconstruction accuracy	r	St. dev.	t	p	Reconstruction accuracy	r	St. dev.	t	p
Time series					Functional connectivity maps				
GSP DCGAN					GSP DCGAN				
Temporal cortex	0.33	0.02	65.49	<0.001	Temporal cortex	0.62	0.06	47.88	<0.001
Lateral frontal cortex	0.16	0.07	9.74	<0.001	Lateral frontal cortex	0.60	0.04	65.04	<0.001
Medial frontal cortex	0.23	0.04	23.24	<0.001	Medial frontal cortex	0.61	0.05	51.87	<0.001
Lateral parietal cortex	0.25	0.03	37.03	<0.001	Lateral parietal cortex	0.70	0.03	110.62	<0.001
Occipital cortex	0.37	0.06	26.65	<0.001	Occipital cortex	0.79	0.05	68.39	<0.001
GSP (no GSR) DCGAN					GSP (no GSR) DCGAN				
Temporal cortex	0.33	0.02	69.29	0.001	Temporal cortex	0.61	0.06	42.19	<0.001
Lateral frontal cortex	0.17	0.08	9.93	<0.001	Lateral frontal cortex	0.62	0.05	59.56	<0.001
Medial frontal cortex	0.23	0.04	23.64	<0.001	Medial frontal cortex	0.60	0.06	46.13	<0.001
Lateral parietal cortex	0.25	0.03	34.43	<0.001	Lateral parietal cortex	0.69	0.03	99.54	<0.001
Occipital cortex	0.37	0.07	25.33	<0.001	Occipital cortex	0.78	0.06	60.96	<0.001
GSP diffusion					GSP diffusion				
Temporal cortex	0.24	0.06	18.20	<0.001	Temporal cortex	0.51	0.10	22.54	<0.001
Lateral frontal cortex	0.10	0.04	9.70	<0.001	Lateral frontal cortex	0.40	0.11	15.85	<0.001
Medial frontal cortex	0.17	0.05	16.11	<0.001	Medial frontal cortex	0.46	0.09	22.03	<0.001
Lateral parietal cortex	0.18	0.04	22.14	<0.001	Lateral parietal cortex	0.52	0.05	47.23	<0.001
Occipital cortex	0.29	0.08	16.46	<0.001	Occipital cortex	0.59	0.13	19.42	<0.001
HCP DCGAN					HCP DCGAN				
Temporal cortex	0.31	0.05	27.60	<0.001	Temporal cortex	0.60	0.07	38.04	<0.001
Lateral frontal cortex	0.12	0.04	14.33	<0.001	Lateral frontal cortex	0.44	0.05	39.47	<0.001
Medial frontal cortex	0.18	0.05	16.73	<0.001	Medial frontal cortex	0.46	0.06	32.53	<0.001
Lateral parietal cortex	0.24	0.05	19.71	<0.001	Lateral parietal cortex	0.62	0.03	81.92	<0.001
Occipital cortex	0.35	0.07	23.39	<0.001	Occipital cortex	0.65	0.06	46.90	<0.001
HCP (temporally smoothed) DCGAN					HCP (temporally smoothed) DCGAN				
Temporal cortex	0.31	0.05	27.32	<0.001	Temporal cortex	0.62	0.06	43.39	<0.001
Lateral frontal cortex	0.14	0.04	15.86	<0.001	Lateral frontal cortex	0.56	0.05	54.81	<0.001
Medial frontal cortex	0.20	0.05	17.41	<0.001	Medial frontal cortex	0.58	0.06	44.94	<0.001
Lateral parietal cortex	0.25	0.05	21.64	<0.001	Lateral parietal cortex	0.68	0.03	97.91	<0.001
Occipital cortex	0.37	0.06	27.52	<0.001	Occipital cortex	0.79	0.06	63.31	<0.001
HCP diffusion					HCP diffusion				
Temporal cortex	0.23	0.05	21.52	<0.001	Temporal cortex	0.47	0.08	26.32	<0.001
Lateral frontal cortex	0.07	0.04	8.44	<0.001	Lateral frontal cortex	0.38	0.06	29.60	<0.001
Medial frontal cortex	0.12	0.05	11.34	<0.001	Medial frontal cortex	0.36	0.07	21.59	<0.001
Lateral parietal cortex	0.17	0.05	14.26	<0.001	Lateral parietal cortex	0.58	0.04	66.37	<0.001
Occipital cortex	0.28	0.07	18.90	<0.001	Occipital cortex	0.63	0.07	39.20	<0.001

R coefficients and standard deviations are shown for correlations between original and reconstructed data. T and p values indicate whether correlations are significantly different from 0.

DCGAN: Deep convolutional generative adversarial networks; GSP: Genomic Superstruct Project; GSR: Global signal regression; HCP: Human Connectome Project.

Figure 4

Fig. 4. DCGAN successfully reconstructs images across datasets, and tSNR of the training data modulates reconstructive accuracy. Box-and-whisker plots are shown, with the center line indicating the median, box limits indicating upper and lower quartiles, whiskers indicating 1.5 times the interquartile range, and plus signs indicating outliers. **(A)** GSP’s reconstructive accuracy is generally greater than HCP’s (time series: $t(30.21) = 2.41$, $p = 0.022$ (marginally significant after Bonferroni correction); FC maps: $t(38) = 12.88$, $p < 0.001$). Its reconstructive accuracy for time series is non-significantly different from the temporally-smoothed (t.s.) HCP’s ($t(32.12) = 1.16$, $p = 0.26$), and its FC map reconstructive accuracy is marginally greater when correcting for multiple comparisons ($t(38) = 2.43$, $p = 0.020$). Temporally-smoothed HCP data yielded more accurate reconstructions than the raw HCP data (time series: $t(19) = -4.67$, $p < 0.001$; FC maps: $t(19) = 143.21$, $p < 0.001$). **(B)** In accordance with these results, the temporal signal-to-noise ratio (tSNR) is higher in the datasets that yielded more accurate reconstructive accuracy. The tSNR of the GSP dataset is greater than both the raw HCP ($t(139.43) = 49.31$, bootstrapped $p = 0.001$) and t.s. HCP ($t(198) = 23.26$, bootstrapped $p = 0.001$) datasets. Temporally smoothing the HCP data significantly increased its tSNR ($t(99) = -40.68$, $p < 0.001$).

*: $p \leq 0.001$

†: marginally significant with Bonferroni-corrected p value

n.s.: non-significant

2) In their response to my comment 1d), where I asked about the performance benefits relative to a simple diffusion methods, the authors give a conceptual response. While this is fair, my comment was trying to encourage the authors to run the diffusion method as a baseline model that they can compare the performance of their neural network to. This evaluation would then allow them to provide an answer that is supported by this assessment, and in particular whether their approach outperforms rather ‘naive’ inpainting methods that simply operate based on tissue blurring.

We agree with the reviewer that testing a more naïve reconstruction method provides a good basis for comparison, and apologize for previously including only a conceptual response. As the reviewer suggested, we performed diffusion-based reconstructions using the GSP data, whereby each compromised vertex was assigned a BOLD value based on the average of the BOLD signal in

adjacent vertices. The process was started at the perimeter of the compromised region so that those vertices could be filled in using the adjacent uncompromised vertices. Then, reconstruction moved inwards, with new vertices being assigned a BOLD value based on the adjacent vertices that were reconstructed in the previous step.

As predicted, both time series and FC maps reconstructed by the diffusion model (see Table 1 for results in each cortical region) were worse than those derived from the DCGAN model (reconstructed time series: paired samples t-test, $t(19)=9.15$, $p < 0.001$; reconstructed FC maps: paired samples t-test, $t(19)=27.62$, $p < 0.001$) (new Fig. 5). In the new Figure 3, we show FC maps generated from i) original, ii) DCGAN-reconstructed, and iii) diffusion-reconstructed vertices in all five cortical areas; DCGAN FC maps are more similar to the original ones than the diffusion FC maps. We replicated this result in the truncated raw HCP dataset (reconstructed time series: paired samples t-test, $t(19)=176.20$, $p < 0.001$; reconstructed FC maps: paired samples t-test, $t(19)=53.50$, $p < 0.001$) (new Fig. 5). This finding suggests that our DCGAN model was able to reconstruct BOLD signals using information beyond the nearby vertices, likely including time series and FC information from distant areas.

We have added the following text and figures to the manuscript:

Results:

“To assess the power of DCGAN, we compared its performance to a simpler diffusion-based method for filling in compromised cortical regions (see *Methods*). The diffusion model was able to reconstruct both time series and FC maps (Table 1). As predicted, its reconstructions are less accurate than the DCGAN model’s (time series: mean difference=0.07, CI [0.06, 0.09], $t(19)=9.15$, $p<0.001$, $\eta^2=0.82$; FC maps: mean difference=0.18, CI [0.16, 0.19], $t(19)=27.62$, $p<0.001$, $\eta^2=0.98$) (Fig. 5). As an example, in Fig. 3 we show FC maps generated from i) original (top row), ii) DCGAN-reconstructed (middle row), and iii) diffusion-reconstructed (bottom row) vertices in all five cortical areas. We replicated these results in the raw HCP dataset (reconstructed time series: mean difference=0.07, CI [0.065, 0.067], $t(19)=176.20$, $p<0.001$, $\eta^2=1.00$; reconstructed FC maps: mean difference=0.06, CI [0.059, 0.064], $t(19)=53.50$, $p<0.001$, $\eta^2=0.99$) (Fig. 5; Table 1). This finding suggests that our DCGAN model extrapolates information embedded within nearby as well as distant cortical regions to reproduce patterns of brain activity, while more naïve methods can only rely on nearby information.”

Discussion:

“The DCGAN model outperformed a more naïve diffusion-based reconstruction method, indicating that machine learning is able to extrapolate information embedded within the whole BOLD image, while simpler filling-in methods are restricted to using nearby information, thereby limiting their ability to capture principles of brain organization.”

Figure 3

Fig. 3. The DCGAN-generated functional connectivity maps are highly similar to the original maps, throughout cortex. The top row shows functional connectivity (FC) maps of seeds from intact BOLD frames in the temporal, lateral frontal, medial frontal, lateral parietal, and occipital cortices. The middle row shows the FC maps of the same seeds extracted from DCGAN-reconstructed BOLD frames. The similarity between the original and reconstructed FC maps is high, with the following correlation coefficients: $r = 0.96$ for the temporal seed, $r = 0.84$ for the lateral frontal seed, $r = 0.89$ for the medial frontal seed, $r = 0.84$ for the lateral parietal seed, and $r = 0.88$ for the occipital seed. The bottom row shows diffusion-reconstructed FC maps. While correlations with original FC maps are also high, they are consistently lower than DCGAN correlations: $r = 0.84$ for the temporal seed, $r = 0.80$ for the lateral frontal seed, $r = 0.42$ for the medial frontal seed, $r = 0.71$ for the lateral parietal seed, and $r = 0.83$ for the occipital seed.

Figure 5

Fig. 5. The DCGAN model outperforms a diffusion-based model. Box-and-whisker plots are shown, with the center line indicating the median, box limits indicating upper and lower quartiles, whiskers indicating 1.5 times the interquartile range, and plus signs indicating outliers. DCGAN-reconstructed time series are more similar to the original ones than diffusion-reconstructed time series, in both the GSP dataset ($t(19) = 9.15, p < 0.001$) and the HCP dataset ($t(19) = 176.20, p < 0.001$). The DCGAN model also outperforms the diffusion model when reconstructing functional connectivity (FC) maps, in both the GSP ($t(19) = 27.62, p < 0.001$) and HCP ($t(19) = 53.50, p < 0.001$) datasets.

*: $p \leq 0.001$

3) My prior question 5) asked about the stability of their algorithm relative to including/omitting GSR. This was not fully answered in the authors response.

We apologize for the incomplete response. We agree that it is important to address the issue of GSR in resting-state FC studies, and we therefore reran our analyses using GSP data preprocessed without GSR. We found that the reconstructive accuracy using these data was very similar to the previous reconstruction results (see Table 1 and new Supplementary Fig. 2). The data processed without GSR yielded significantly lower reconstructive accuracy for time series (paired samples t-test: $t(19) = -2.81, p = 0.01$) and significantly higher reconstructive accuracy for FC maps (paired samples t-test, $t(19) = 4.87, p < 0.001$) compared to GSR-processed data. However, the effect size was negligible in both cases, as evidenced by the extremely low mean differences in r coefficients: -0.001 for time series and 0.003 for FC maps. This finding indicates that GSR does not affect machine learning reconstructive accuracy in any meaningful way, and that the information learned by the model is stable.

We have added the following text and figures to the manuscript:

Results:

“The resting-state fMRI data used for reconstruction were preprocessed with global signal regression (GSR), which introduces spurious temporal anticorrelations in FC analysis¹⁴. While progress has

been made, there is still no consensus about whether GSR should or should not be included in resting-state data preprocessing¹⁵. To ensure the robustness of our results, we retrained our DCGAN model using the same data, albeit preprocessed without GSR. This data yielded lower reconstructive accuracy for time series (mean difference=-0.001, CI [-0.003, -0.0004], $t(19)=-2.81$, $p=0.01$, $\eta^2=0.29$) and higher reconstructive accuracy for FC maps (mean difference=0.003, CI [0.002, 0.005], $t(19)=4.87$, $p<0.001$, $\eta^2=0.55$) compared to GSR-preprocessed data (Supplementary Fig. 2; see Table 1 for reconstruction details on each cortical region). Importantly, however, the effect was negligible in both cases, as evidenced by the near-zero mean differences in r coefficients: -0.001 (CI [-0.003, -0.0004]) for time series and 0.003 (CI [0.002, 0.005]) for FC maps (Supplementary Fig. 2). This finding indicates that GSR does not affect machine learning reconstructive accuracy in any meaningful way, and that the information learned by the DCGAN model is stable. The remaining analyses were conducted on GSR-preprocessed GSP data.”

Discussion:

“Additional control analyses revealed that global signal regression (GSR), a preprocessing step that amplifies anti-correlations in the brain through its mathematical mandate^{14,15}, does not meaningfully impact the learning of these principles.”

Supplementary Figure 2

Supplementary Fig. 2. Reconstructive accuracy is stable regardless of whether global signal regression is applied to the data. Compared to data preprocessed with global signal regression (GSR), data preprocessed without GSR yielded significantly lower reconstructive accuracy for time series ($t(19) = -2.81$, $p = 0.01$) and significantly higher reconstructive accuracy for FC maps ($t(19) = 4.87$, $p < 0.001$). Crucially, the effect is so small (mean differences in r coefficients: -0.001 for time series and 0.003 for FC maps) that it is deemed inconsequential. GSR therefore has no meaningful impact on DCGAN reconstructive accuracy. Error bars represent standard errors.

*: $p \leq 0.05$

** : $p \leq 0.001$

References

Triantafyllou, C., Hoge, R. D., Krueger, G., Wiggins, C. J., Potthast, A., Wiggins, G. C., & Wald, L. L. (2005). Comparison of physiological noise at 1.5 T, 3 T and 7 T and optimization of fMRI acquisition parameters. *Neuroimage*, 26(1), 243-250.

Marcus, D. S., Harms, M. P., Snyder, A. Z., Jenkinson, M., Wilson, J. A., Glasser, M. F., ... & Hodge, M. (2013). Human Connectome Project informatics: quality control, database services, and data visualization. *Neuroimage*, 80, 202-219.

REVIEWERS' COMMENTS

Reviewer #2 (Remarks to the Author):

I thanks the authors for the additional revisions, and am now happy with the manuscript.